# LanguageBind: Extending Video-Language Pretraining to N-modality by Language-based Semantic Alignment

**Bin Zhu[1,3,*], Bin Lin[1,*], Munan Ning[1,2], Yang Yan[1], JiaXi Cui[1], Hongfa Wang[3], Yatian Pang[4], Wenhao Jiang[6], Junwu Zhang[1], Zongwei Li[3], Wancai Zhang[5], Zhifeng Li[3], Wei Liu[3], Li Yuan[1,2,†]**

[1]Peking University, [2]Pengcheng Lab, [3]Tencent Data Platform, [4]National University of Singapore, [5]Nari Technology Development Limited Company, [6]Guangdong Laboratory of Artificial Intelligence and Digital Economy (SZ)

{binzhu,linbin.ece}@stu.pku.edu.cn

## Abstract

The video-language (VL) pretraining has achieved remarkable improvement in multiple downstream tasks. However, the current VL pretraining framework is hard to extend to multiple modalities (N modalities, $N \geq 3$) beyond vision and language. We thus propose ***LanguageBind***, taking the language as the bind across different modalities because the language modality is well-explored and contains rich semantics. Specifically, we freeze the language encoder acquired by VL pretraining and then train encoders for other modalities with contrastive learning. As a result, all modalities are mapped to a shared feature space, implementing multi-modal semantic alignment. While LanguageBind ensures that we can extend VL modalities to N modalities, we also need a high-quality dataset with alignment data pairs centered on language. We thus propose ***VIDAL-10M*** with 10 ***M***illion data with ***V***ideo, ***I***nfrared, ***D***epth, ***A***udio and their corresponding ***L***anguage. In our VIDAL-10M, all videos are from short video platforms with complete semantics rather than truncated segments from long videos, and all the video, depth, infrared, and audio modalities are aligned to their textual descriptions. LanguageBind has achieved superior performance on a wide range of 15 benchmarks covering video, audio, depth, and infrared. Moreover, multiple experiments have provided evidence for the effectiveness of LanguageBind in achieving indirect alignment and complementarity among diverse modalities.

## 1 Introduction

With the development of the Internet and smartphones, there has been a proliferation of video websites and apps (*e.g.*, Youtube and TikTok), leading to a substantial increase number of videos (Xue et al., 2022). Consequently, a set of video tasks have emerged, such as video search (Smith & Chang, 1997), video recommendation (Deldjoo et al., 2016), and video editing (Casares et al., 2002; Bonneel et al., 2014). To solve video understanding tasks, video-language pretraining has been employed by training foundation models by combining computer vision (He et al., 2016; Dosovitskiy et al., 2020) and natural language processing (Vaswani et al., 2017). These models can capture video semantics and solve downstream tasks (Karpathy et al., 2014; Mithun et al., 2018).

However, current VL pretraining frameworks are often limited to vision and language modalities. The ImageBind (Girdhar et al., 2023) introduces an indirect alignment method for multi-modal pretraining. It aligns other modalities to images, facilitating a comprehensive understanding of various modalities such as infrared (Jia et al., 2021), depth (Kim et al., 2022), audio (Piczak, 2015), and IMU (Grauman et al., 2022). In practical tasks such as zero-shot retrieval and classification

---

*Equal contribution.
†Corresponding author.

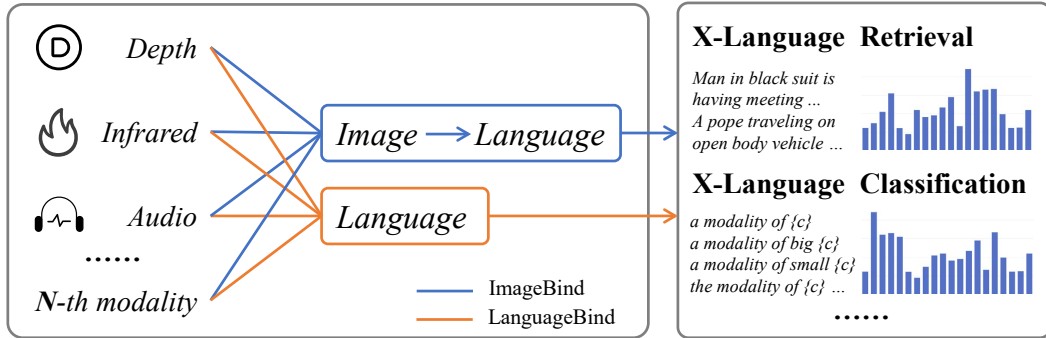

Figure 1: **ImageBind vs. LanguageBind**. The ImageBind method relies on images as intermediaries, while the LanguageBind method dispenses with this requirement. LanguageBind directly aligns all modalities to the language space, thereby enhancing its applicability to downstream tasks. "X" represents all modalities except language, and "c" represents category.

as shown in Figure 1, the alignment with language modality is predominantly required for various modalities. While the indirect alignment of ImageBind may result in performance degradation, the LanguageBind method does not need images as intermediaries and facilitates straightforward expansion to additional modalities in downstream tasks.

In this paper, we propose the ***LanguageBind***, a language-based multi-modal pretraining framework that can extend video-language pretraining to multiple (N) modalities. As the language modality contains rich semantic information and is well-explored (Kenton & Toutanova, 2019; Dai et al., 2019), we take it as the bind across different modalities. This process maps all modalities to a unified embedding space, enabling effective semantic alignment. To improve training efficiency, we employ Low-Rank Adaptation (LoRA) (Hu et al., 2021) for fine-tuning, achieving impressive training results with minimal training iterations.

To further improve the modal integrity in pretraining and validate our LanguageBind, we introduce a dataset with five modalities, the ***VIDAL-10M***, which includes VL, IL (infrared-language), DL (depth-language), and AL (audio-language) data pairs. The videos of previous datasets are always truncated segments from long videos (Miech et al., 2019; Xue et al., 2022), resulting in fragmented semantics. To avoid this problem, we construct our video-text pairs from short videos with complete stories. To ensure the quality of the central language modality, we perform multi-view text generation and enhancement on VIDAL-10M.

The proposed LanguageBind ensures that we can extend vision-language to multiple (N) modalities, and our dataset VIDAL-10M benefits more downstream tasks beyond VL tasks, including video retrieval (Luo et al., 2022), depth classification (Cao et al., 2017), infrared classification (Baffa & Lattari, 2018) and audio classification (Palanisamy et al., 2020). As shown in Figure 2, LanguageBind achieves superior performances on a broad range of 15 tasks. In zero-shot text to video retrieval, LanguageBind achieves superior performance on four datasets, surpassing InterVideo (Wang et al., 2022c) by 1.9% on MSR-VTT (Xu et al., 2016), 8.8% on MSVD (Chen & Dolan, 2011), 6.3% on DiDeMo (Anne Hendricks et al., 2017), and 4.4%

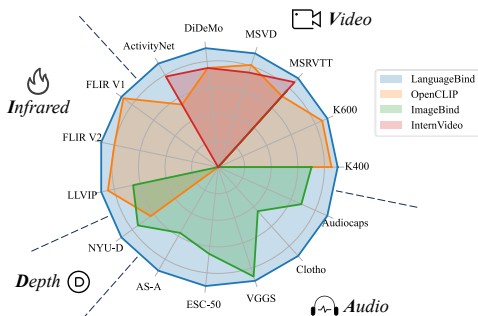

Figure 2: LanguageBind achieves superior performances on **15 benchmarks.**

on ActivityNet (Caba Heilbron et al., 2015). For zero-shot classification on depth and infrared data, LanguageBind achieves a substantial performance advantage over ImageBind. LanguageBind attains top-1 accuracy of 87.2% and 65.1% on LLVIP and NYU-D, respectively, outperforming ImageBind by 23.8% and 11.1%. For zero-shot audio classification tasks, LanguageBind outperforms ImageBind with a 23.8% higher top-1 accuracy on the ESC50 dataset.

**We summarize our primary contributions as follows**:

- We propose *LanguageBind*, the langauge-based multi-modal pretraining approach. During the pretraining process, all modalities gradually align with the language modality through contrastive learning, and these modalities are unified within a shared embedding space.

- We introduce *VIDAL-10M*, a large-scale five-modal video dataset, containing 10 million data pairs with aligned VL, IL, DL, and AL. To the best of our knowledge, *VIDAL-10M* is the first large-scale video dataset with depth and infrared modalities.

- Extensive experiments validate the effectiveness of our dataset and approach, achieving remarkable performance in video and other modality understanding tasks.

## 2 RELATED WORK

**Multi-modal Pretraining** Multi-modal pretraining begins with pretraining in vision and language. CLIP (Radford et al., 2021) pioneered the alignment of images and texts on a large-scale dataset comprising 400 million samples, effectively establishing a bridge between the image and text domains. This alignment benefits a variety of downstream tasks, including zero-shot classification and image-text retrieval (Li et al., 2023a). CLIP can also be used as a foundation for alignment in other modalities. For instance, CLIP4Clip (Luo et al., 2022) aligns video with text, CLAP (Wu* et al., 2023) aligns audio with text, and PointCLIP (Zhang et al., 2022) aligns point clouds with text. Recent efforts have undertaken a comprehensive exploration of multi-modal alignment through pretraining. Augmenting the alignment process with additional modalities can enhance the model's robustness while maintaining its performance, as observed in VALOR (Chen et al., 2023a) and VAST (Chen et al., 2023b). However, as the number of modalities increases, the training paradigm required to align them effectively undergoes significant changes. Meta-transformer (Zhang et al., 2023) accommodates 12 modalities and utilizes distinct tokenizers to harmonize the embedding space across modalities. ImageBind (Girdhar et al., 2023) expands multi-modal alignment pretraining to encompass six modalities but may not perform as well in language-related tasks due to indirect alignment. In our work, we propose LanguageBind, a direct alignment mechanism designed to align alternative modalities directly with the language modality, which has the highest information density. This direct alignment mechanism yields discernible improvements in downstream task performance.

**Multi-modal Datasets** Multi-modal datasets serve as the foundation for multi-modal pretraining (Zhu et al., 2024; Lin et al., 2023). Initially, these datasets only consisted of videos and their corresponding categories, as shown in Table 1. HMDB-51 (Kuehne et al., 2011) and UCF-101 (Soomro et al., 2012) are examples of such datasets, which contain truncated segments from long videos with manual annotation. However, creating these datasets required significant human effort, which limited their scalability and diversity. To address this issue, researchers turned their attention to the abundance of video-text resources available on the internet. Inspired by the success of image-text datasets (Sharma et al., 2018; Changpinyo et al., 2021), they used script-based programming (Schuldt et al., 2004; Kong et al., 2019; Sigurdsson et al., 2018) to extract millions of video-text data pairs. However, acquiring data from modalities like infrared (Teledyne FLIR, 2015a;b) and depth (Silberman et al., 2012), which require spe-

Table 1: Comparision of existing multi-modal datasets. VIDAL-10M is the currently first accessible multi-modal dataset including aligned VL, IL, DL, and AL data pairs.

| Datasets | Samples | Modality | Year |
|---|---|---|---|
| HMDB-51 | 7K | V | 2011 |
| UCF-101 | 13K | V | 2012 |
| ActivityNet-200 | 20K | VT | 2015 |
| WebVid-10M | 10.7M | VT | 2021 |
| HD-VILA-100M | 100M | VT | 2022 |
| HowTo-100M | 136M | VT | 2019 |
| LLVIP | 15k | VI | 2021 |
| FLIR V1 | 10k | VI | 2015 |
| FLIR V2 | 12k | VI | 2015 |
| NYU-D | 1.4k | VD | 2012 |
| YouTube-8M | 6.1M | VAT | 2016 |
| AVA | 58K | VAT | 2017 |
| **VIDAL-10M (Ours)** | 10M | VIDAL | 2023 |

cialized equipment and manual annotation, has been challenging. This has severely limited the scale of the data and its alignment with other modalities. Although existing work like ImageBind (Girdhar et al., 2023) has attempted to bind various image-paired datasets and achieve indirect semantic alignment between different modalities, this approach still faces issues of incomplete and indirect data alignment. Thus, there is an urgent need for multi-modal datasets with direct semantic aligned data pairs, especially for modalities with five or more types.

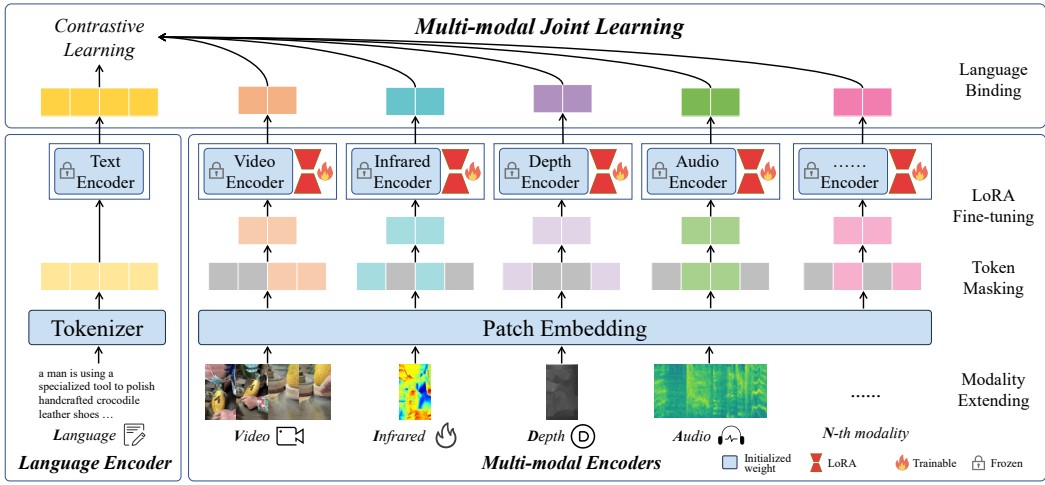

Figure 3: **LanguageBind overview**. The language encoder parameters are frozen, while the multi-modal encoder parameters can be adjusted using the LoRA technique. By employing contrastive learning between language and other modalities, LanguageBind successfully achieved multimodal joint learning, thereby fostering semantic alignment across different modalities.

## 3 METHOD

In this section, we present LanguageBind, a multi-modal pretraining approach designed to align the semantics of different modalities and enhance cross-modal retrieval and zero-shot classification. As shown in Figure 3, LanguageBind consists of three parts: (a) multi-modal encoders, (b) language encoder, and (c) multi-modal joint learning.

### 3.1 MULTI-MODAL ENCODERS

For other modalities besides language, we employ the 24-layer, 1024-dimensional vision transformer with a patch size of 14. The encoders are initialized from OpenCLIP-large (Ilharco et al., 2021). Depth and infrared are treated as RGB images, which are replicated 3 times in the channel dimension to align with RGB images. Following ImageBind, audio data is transformed into spectrograms with a duration of 10 seconds (128 mel-bins) and we repeat and pad the spectrograms. For example, a 4-second spectrogram would be repeated twice and then padded with zero for an additional 2 seconds. Similarly, we also replicate it 3 times in the channel dimension. If the duration exceeds 10 seconds, we randomly sample three 10-second audio segments, each from the front 1/3, middle 1/3, and back 1/3 of the original audio, and finally stack them together.

**Patch masking** To address the inefficiency in processing all tokens within the encoder, we divide the image into patches and take a small portion of patches by encoder mask $\mathbb{M}_e$, following MAE (He et al., 2022). Given a modality $\boldsymbol{m} \in \mathbb{R}^{H \times W \times C}$, where $(H, W)$ represents the resolution of the original data, with $C$ denoting the number of channels. We first transform it into patches using a patch embedding layer with non-overlapping filters. This operation produces patches denoted as $\boldsymbol{m}' \in \mathbb{R}^{N \times C}$ and $N = \frac{H \times W}{S^2}$ represents the resulting sequence length, where $S$ represents the size of each patch. Subsequently, positional embedding is applied to the visible tokens, which are divided by encoder mask. The combined sequence $\boldsymbol{x}$ is defined as:

$$\boldsymbol{x} = \{\boldsymbol{m}'_i + \boldsymbol{P}_i\}_{i \in \mathbb{M}_e} \tag{1}$$

where $\boldsymbol{P}$ is a sequence of learnable position tokens, and $i$ represents the position index at patches.

**LoRA fine-tuning** We employ the LoRA technique (Hu et al., 2021) to accelerate fine-tuning. For a modality-agnostic encoder with a weight matrix $W_0 \in \mathbb{R}^{d \times k}$, we maintain the weight matrix $W_0$ frozen while learning a new weight matrix $BA$. For instance, in the case of the modality-agnostic encoder $h(\cdot)$ and $\boldsymbol{x}$, the forward process can be represented as follows:

$$h(\boldsymbol{x}) = W_0 \boldsymbol{x} + BA\boldsymbol{x} \tag{2}$$

where $B \in \mathbb{R}^{d \times r}, A \in \mathbb{R}^{r \times k}$, with $r$ being the minimum of $d$ and $k$. It is important to highlight that both $W_0$ and $BA$ possess the same input and output dimensions, facilitating their summation to produce the final output. **Modality extending** To extend the LanguageBind method to multiple (N) modalities, the first step involves processing the data into a sequence of tokens. Subsequently, the parameters are initialized from OpenCLIP. The encoder for different modalities is then trained through token masking and LoRA fine-tuning while keeping the language encoder frozen. Finally, this modality is aligned with the language feature space.

## 3.2 LANGUAGE ENCODER AND MULTI-MODAL JOINT LEARNING

For the language encoder, we utilize a 12-layer transformer model with 768-dimensional and initialize it from OpenCLIP. For a given text, we initially employ a BPE tokenizer to segment words into relatively common subwords. Each subword corresponds to a unique token, and these tokens are embedded within a word embedding layer. Ultimately, the tokens are encoded by the language encoder to obtain a text logit $y \in \mathbb{R}^{L \times C}$, where $L$ represents the length of the sequence. To ensure alignment across different modalities, we implement contrastive learning principles (Radford et al., 2021). The objective of this approach is to increase the similarity of paired data, bringing them closer to the same semantic space, while minimizing the similarity of unpaired data. We utilize contrastive learning to bind individual modalities to language.

$$L_{M2T} = -\frac{1}{K} \sum_{i=1}^{K} \log \frac{\exp(x_i^\top y_i / \tau)}{\sum_{j=1}^{K} \exp(x_i^\top y_j / \tau)}, L_{T2M} = -\frac{1}{K} \sum_{i=1}^{K} \log \frac{\exp(y_i^\top x_i / \tau)}{\sum_{j=1}^{K} \exp(y_i^\top x_j / \tau)} \quad (3)$$

where $x_i$ is the $i$-th modality data and $y_j$ is the $j$-th text and their features are normalized. $K$ and $\tau$ are batch size and the temperature. The direct alignment of each modality **M** with language **T** enables us to significantly enhance zero-shot classification and retrieval tasks.

## 4 THE VIDAL-10M DATASET

In this section, we will describe how to construct our VIDAL-10M dataset, including 3 million pairs of video-language data, 3 million pairs of infrared-language data, 3 million pairs of depth-language data, and 1 million pairs of audio-language data. As shown in Figure 4, the collection process consists of three main steps: visual search term database construction (Section 4.1), video and audio collection and filtering (Section 4.2), and modality generation and enhancement (Section 4.3).

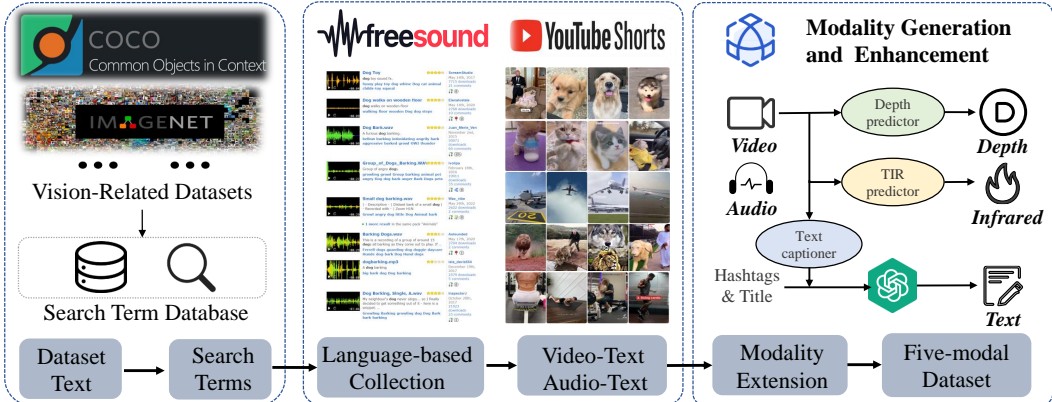

Figure 4: **VIDAL-10M construction.** (a) Firstly, a search term database is generated by leveraging visually related datasets. (b) Subsequently, relevant videos and audios are collected from the internet and undergo a series of filtering processes. (c) Lastly, we perform infrared and depth modality generation, as well as multi-view text generation and enhancement.

## 4.1 VISUAL SEARCH TERM DATABASE CONSTRUCTION

To build a video dataset with rich visual concepts and diversity, we design a unique search term acquisition strategy. We leverage text data including labels and captions from various visual task

datasets (YouTube-8M (Abu-El-Haija et al., 2016), MSR-VTT (Xu et al., 2016), COCO (Lin et al., 2014), AVA (Gu et al., 2018), HMDB-51 (Kuehne et al., 2011), ImageNet (Deng et al., 2009)) to create a large-scale search term database with diversity and broad applicability. Then we filter these search terms based on their frequency and employ the NLTK toolkit for part-of-speech tagging, followed by tallying the occurrences of keywords (nouns and verbs). A balanced subset of 100,000 search items corresponding to these keywords is then extracted as the final search team database.

## 4.2 VIDEO AND AUDIO COLLECTION AND FILTERING

During the data collection process, we utilize the aforementioned search terms to retrieve video-text pairs and audio-text pairs from relevant platforms, e.g. YouTube Shorts, and Freesound. Regarding video collection, in order to obtain short videos with high-quality textual descriptions, we implemented a filtering mechanism for the title and hashtags. Video samples with titles containing less than 2 words and without video hashtag labels are excluded from our dataset. Moreover, we removed irrelevant words and hashtags, such as "youtube", "fyp", "shorts", etc. Furthermore, to ensure a complete, consistent, and precise depiction of the event within a single full video, we decide to impose a duration limit of 20 seconds. Shorter videos tend to exhibit better scene coherence and event integrity and are more closely aligned with corresponding hashtags and title descriptions. Ultimately, we obtain a short video dataset that encompasses more specific rather than abstract content. Concerning audio collection, we rank the audio list on different audio platforms based on its similarity to the search terms. Additionally, we conduct filtering operations similar to those for videos, taking into account factors such as audio ratings, download counts, user comments, tags, and duration. This comprehensive approach allows us to curate and refine the audio and video content more effectively.

## 4.3 MODALITY GENERATION AND ENHANCEMENT

**Multi-view text generation and enhancement** The language modality of VIDAL-10M consists of multi-view texts, including title, hashtags, keyframe captions, video captions, and enhanced captions. The detailed text generation and enhancement pipeline is illustrated in the Appendix, section A. Hashtags in VIDAL-10M are specifically designed to highlight the main subjects and actions depicted in the video. These hashtags serve as key indicators, emphasizing the focal points and dynamic elements of the video. However, hashtags alone may not fully capture the spatial information conveyed by the video frames. To address this limitation, we leverage the image captioning model OFA (Wang et al., 2022b) to generate supplementary keyframe captions that enrich the spatial information at the keyframe level. These captions also contain local temporal information related to the video content, which is beneficial for visual-text pretraining. Besides spatial information, temporal information concealed within the video is equally significant, providing crucial insights into the progression and sequencing of events within the video. To further supplement the overall thematic and temporal information of the video, we employ the mPLUG-owl model (Ye et al., 2023) to generate video captions based on the combination of video, title, and hashtags. By leveraging the title and hashtags as accurate video labels, we guide the mPLUG-owl model to generate captions that align with the video theme, reducing potential model bias to a certain extent. Furthermore, to extract valuable information from the generated video captions, we utilize the ChatGPT model to refine and enhance the textual description, thereby greatly improving the quality of the text. By incorporating the above text components, multi-view textual descriptions provide a comprehensive and detailed representation of the video content.

**Infrared and depth modality generation** In the field of depth and infrared, creating modal datasets typically requires specialized equipment and human effort, resulting in limited data. Despite the success of large-scale pretraining models (Radford et al., 2021; Wu* et al., 2023; Luo et al., 2022; Chen et al., 2023b) in NLP and CV, there remains a lack of large-scale data in this field. To address this challenge, we propose using advanced generative models specifically to construct a large-scale dataset of depth and infrared. The sRGB-TIR model (Lee et al., 2023) is used for infrared modality generation and the GLPN model (Kim et al., 2022) for depth modality generation, generating depth and infrared from keyframes in our videos. While some limitations may exist, our collection of millions of video frames and corresponding texts with highly diverse semantics can significantly reduce the presence of model biases.

## 5 EXPERIMENTS AND RESULTS

In this section, we evaluate the effectiveness of LanguageBind in various downstream tasks. Firstly, LanguageBind's capability to align video and text is assessed using zero-shot video-text retrieval. Additionally, we use LanguageBind to enhance the performance of downstream tasks that involve depth, infrared images, and audios. Finally, we conduct ablation experiments to analyze the impact of different parameter configurations and text descriptions on LanguageBind's performance.

### 5.1 ZERO-SHOT RETRIEVAL IN VIDEO-LANGUAGE

**Comparison to prior methods** In the zero-shot text to video retrieval benchmark, we utilize ViT-L/14 as the video encoder and add temporal attention layers for fair comparison, which can be found in Appendix B. According to the results presented in Table 2, the performance of LanguageBind exceeds that of VideoCoca (Yan et al., 2022) and OmniVL (Wang et al., 2022a) by 8.3% and 8.0% respectively on MSR-VTT. In comparison to the ImageBind model utilizing the Vit-Huge architecture, the LanguageBind model, employing the Vit-Large model, showcases superior experimental outcomes. Furthermore, compared to models based on CLIP-Large but using more training data, LanguageBind achieves superior performance on four datasets, outperforming InterVideo (Wang et al., 2022c) by 1.9% on MSR-VTT, 8.8% on MSVD, 6.3% on DiDeMo, and 4.4% on ActivityNet. We also exceed TVTSv2 (Zeng et al., 2023) by 4.4% and 3.2% on MSR-VTT and DiDeMo, respectively. Moreover, we outperforms UMT-L Li et al. (2023b) on all datasets. For a fair comparison of dataset validity, we use the Vit-B/32 model of CLIP4CLIP to conduct validation experiments using the 100K subset of VIDAL-10M and the 380k subset of HowTo100M. As shown in Table2, the VIDAL-100k outperforms the HT100M-380k on both MSRVTT and MSVD datasets, validating the effectiveness of our dataset.

Table 2: **Zero-shot text to video retrieval performance** of LanguageBind across four datasets.

| Method | Dataset | MSR-VTT | | | MSVD | | | DiDeMo | | | ActivityNet | | |
|---|---|---|---|---|---|---|---|---|---|---|---|---|---|
| | | R@1 | R@5 | R@10 | R@1 | R@5 | R@10 | R@1 | R@5 | R@10 | R@1 | R@5 | R@10 |
| *Non-CLIP models* | | | | | | | | | | | | | |
| OmniVL | 14M | 34.6 | 58.4 | 66.6 | - | - | - | 33.3 | 58.7 | 68.5 | - | - | - |
| VideoCoCa | 100M | 34.3 | 57.8 | 67.0 | - | - | - | - | - | - | 34.5 | 63.2 | 76.6 |
| *CLIP-H/14* | | | | | | | | | | | | | |
| ImageBind | - | 36.8 | 61.8 | 70.0 | - | - | - | - | - | - | - | - | - |
| *CLIP-L/14* | | | | | | | | | | | | | |
| UMT | 5M | 33.3 | 58.1 | 66.7 | 44.4 | 73.3 | 82.4 | 34.0 | 60.4 | 68.7 | 31.9 | 69.2 | 72.0 |
| TVTSv2 | 8.5M | 38.2 | 62.4 | 73.2 | - | - | - | 34.6 | 61.9 | 71.5 | - | - | - |
| InternVideo | 12.8M | 40.7 | - | - | 43.4 | - | - | 31.5 | - | - | 30.7 | - | - |
| **LanguageBind** | **3M** | **42.6** | **65.4** | **75.5** | **52.2** | **79.4** | **87.3** | **37.8** | **63.2** | **73.4** | **35.1** | **63.4** | **76.6** |

Table 3: **Zero-shot text to video retrieval** to verify the effectiveness of our dataset.

| Dataset | Method | Parameter | Source | R@1↑ | R@5↑ | R@10↑ | MR↓ |
|---|---|---|---|---|---|---|---|
| MSR-VTT | CLIP4Clip | 86M | WIT400M, HT100M-380k | 32.0 | 57.0 | 66.9 | 4.0 |
| | CLIP4Clip | **86M** | WIT400M, **VIDAL-100k** | **35.7** | **60.8** | **71.5** | **3.0** |
| MSVD | CLIP4Clip | 86M | WIT400M, HT100M-380k | 38.5 | 66.9 | 76.8 | 2.0 |
| | CLIP4Clip | **86M** | WIT400M, **VIDAL-100k** | **42.0** | **70.0** | **79.2** | **2.0** |

### 5.2 ZERO-SHOT IN MULTIPLE MODALITIES

**Zero-shot X-Language classification** We compare our model with the recent state-of-the-art multimodal pretraining models, OpenCLIP (Ilharco et al., 2021) and ImageBind (Girdhar et al., 2023) on multi-modal understanding ability tasks in Table 4. For video zero-shot classification, we outperform ImageBind by 14.0% with a smaller model on Kinetics-400 (Kay et al., 2017), and we also report the results of multi-view/crop (Simonyan & Zisserman, 2014) on OpenCLIP for further comparison. For infrared, LanguageBind exhibits a noteworthy 23.8% performance advantage over

ImageBind on the LLVIP and outperforms OpenCLIP on all three datasets (LLVIP, FLIR V1, and V2). For depth images, our zero-shot results on NYU-D surpass ImageBind by a substantial margin of 11.1% and outperform OpenCLIP by 19.7%. For audio, we outperform ImageBind by 10.1% on Audioset dataset and 1.1% on VGGSound dataset. We outperform ImageBind by a large margin of 23.9% on the ESC-50 dataset.

Table 4: **Zero-shot X-Language classification**. We present the top-1 accuracy results for all datasets, where AS-A represents Audioset Audio-only (Gemmeke et al., 2017).

| Method | Size | Video | | Infrared | | | Depth | Audio | | |
|---|---|---|---|---|---|---|---|---|---|---|
| | | K400 | K600 | LLVIP | FLIR V1 | FLIR V2 | NYU-D | AS-A | ESC-50 | VGGS |
| ImageBind | Huge | 50.0 | - | 63.4 | - | - | 54.0 | 17.6 | 66.9 | 27.8 |
| OpenCLIP | Large | 60.7 | 59.0 | 82.2 | 81.2 | 42.6 | 45.4 | - | - | - |
| **LanguageBind** | **Large** | **64.0** | **61.9** | **87.2** | **82.9** | **48.0** | **65.1** | **27.7** | **91.8** | **28.9** |

**Zero-shot Audio-Language retrieval** We compare zero-shot text-to-audio retrieval on Clotho and Audiocaps datasets. LanguageBind outperformes AVFIC (Nagrani et al., 2022) and ImageBind by margins of 9.1% and 6.1% on Clotho and by 2.9% and 5.5% on Audiocaps, respectively. Moreover, LanguageBind surpasses the powerful baseline of VALOR (Chen et al., 2023a) by 3.7% on the Clotho dataset.

Table 5: **Zero-shot Audio-Language retrieval**

| Method | Clotho | | Audiocaps | |
|---|---|---|---|---|
| | R@1 | R@10 | R@1 | R@10 |
| AVFIC | 3.0 | 17.5 | 8.7 | 37.7 |
| ImageBind | 6.0 | 28.4 | 9.3 | 42.3 |
| VALOR | 8.4 | - | - | - |
| **LanguageBind** | **12.1** | **44.0** | **12.2** | **53.2** |

**Zeor-shot langauge-based multi-modal joint retrieval** In Table 6, we conduct multi-modal joint retrieval to explore the complementarity of joint space. We report the R@1 scores on MSR-VTT and Place datasets, while reporting accuracy on other datasets. For MSR-VTT, we only evaluate using videos that include audio. Integrating audio embeddings for video-language retrieval further improves performance, increasing it from 41.4 to 42.0. Similar trends have been observed in other modalities, where each modality has the potential to enhance the performance when combined with other modalities. These results demonstrate that LanguageBind is capable of learning a more consistent feature space.

**Emergent zero-shot retrieval** As shown in Table 7, we explore the zero-shot performance of emergency coverage in four datasets, including RGB images, audio, infrared, and depth. Due to the novelty of our approach, there are no "fair" baseline models for comparison. Nonetheless, we compare our results with ImageBind, which aligns with images directly. For example, we achieved R@1 scores of 10.6 and 10.0 on AVE (Tian et al., 2018) and VGGS, respectively. On each benchmark, the performance of emergency zero-shot retrieval achieves significant gains, even approaching results obtained by incorporating textual features. These results suggest that LanguageBind aligns various modalities and implicitly transfers text supervision associated with specific modalities and tasks.

Table 6: **Zeor-shot langauge-based multi-modal joint retrieval.** * donates that it is not clear whether only videos with audio are included. † donates that dark nighttime images.

| Dataset | Method | Task | Top-1 |
|---|---|---|---|
| MSR | ImageBind | V→T | 36.1* |
| | | A+V→T | 36.8 (+0.7) |
| | Ours | V→T | 41.4 |
| | | A+V→T | 42.0 (+0.6) |
| NYU | ImageBind | D→T | 54.0 |
| | Ours | D→T | 65.1 |
| | | RGB→T | 76.0 |
| | | D+RGB→T | 77.4 (+1.4) |
| LLVIP | Ours | RGB†→T | 62.4 |
| | | I+RGB†→T | 79.3 (+16.9) |

Table 7: **Emergent zero-shot retrieval.** † donates that we randomly select 10% data to test.

| Dataset | Method | Task | Emergent | R@1 |
|---|---|---|---|---|
| AVE† | Ours | V→A | ✔ | 10.6 |
| | ImageBind | | ✗ | 36.9 |
| VGGS† | Ours | V→A | ✔ | 10.0 |
| | ImageBind | | ✗ | 28.7 |
| LLVIP† | Ours | RGB→I | ✔ | 7.5 |
| | | RGB+T→I | ✗ | 9.1 |
| | | I→RGB | ✔ | 9.3 |
| | | D+I→RGB | ✗ | 10.6 |
| NYU | Ours | RGB→D | ✔ | 17.9 |
| | | RGB+T→D | ✗ | 18.3 |
| | | D→RGB | ✔ | 24.5 |
| | | D+T→RGB | ✗ | 25.7 |

## 5.3 TRAINING LOSS AND ARCHITECTURE

Following ImageBind, we mainly focus on depth and infrared, which are visual and spatial modality. We report R@1 score for Clotho and top-1 accuracy for others.

**Training epochs.** We conduct an experiment in Table 8a to study the effect of training epochs which shows that the LoRA fine-tuning is highly effective. Although 3 epochs of training regimen yield superior accuracy, we chose to optimize for a single epoch, achieving a balance between performance and training cost.

**Training batch size.** In Table 8b, we evaluate the effect of batch size on representation learning. The experiments have shown that a larger batch size is not necessarily better.

**Training strategy.** In Table 8c, we compare three different strategies. Training from scratch exhibits the poorest performance, likely due to the lack of prior knowledge from CLIP pretraining. On the other hand, full tuning shows significant improvement compared to training from scratch. This highlights the positive impact of leveraging prior knowledge in the form of pre-trained weights. Meanwhile, the LoRA method stands out for its advantages in terms of time and memory cost. It requires less time and memory resources compared to full tuning. Furthermore, LoRA outperforms full tuning on multiple datasets such as LLVIP, FLIRv1, and Clotho. This indicates that LoRA is not only efficient but also effective in learning new knowledge specific to different domains while better retaining the previously acquired knowledge from the pre-trained OpenCLIP models.

**Rank of LoRA.** As detailed in Table 8d. We observe that smaller rank values lead to more significant performance improvements, whereas larger rank tends to decrease performance. This trend may be attributed to the potential overfitting of the model.

**Temperature for loss.** As shown in Table 8e, we find that the learnable temperature initiated from 0.07 performs best, outperforming the fixed temperature strategy proposed by ImageBind.

**Masked ratio.** We explore the impact of different mask ratios in Table 8f. The results show that a mask ratio of 0.5 demonstrates the highest performance, requiring only a quarter of the computational resources, aligning with findings in FLIP (Li et al., 2023c).

Table 8: **Training loss and architecture** design decisions and their impact on zero-shot classification. Settings for results in Section 5.2 highlighted in ⬛ gray .

(a) **Training epochs**

| Dataset | 1 | 5 |
|---------|------|------|
| NYU-D | **65.1** | 64.5 |
| LLVIP | **83.9** | 81.1 |
| FLIR V1 | 82.9 | **85.0** |
| FLIR V2 | **48.0** | 44.7 |

(b) **Training batch size**

| Dataset | 512 | 1k | 2k |
|---------|------|------|------|
| NYU-D | 63.9 | **65.1** | 64.5 |
| LLVIP | 80.0 | **83.9** | 78.6 |
| FLIR V1 | 81.6 | 82.9 | **85.2** |
| FLIR V2 | 45.1 | **48.0** | 47.9 |

(c) **Training strategy**

| | Scratch | Full tuning | LoRA |
|---------|---------|-------------|--------|
| Time | 1.4h | 1.4h | **0.8h** |
| Mems | 278M | 278M | **132M** |
| LLVIP | 57.1 | 85.1 | **87.2** |
| FLIR V1 | 74.7 | 81.3 | **81.6** |
| ESC-50 | 86.8 | **88.9** | 87.4 |
| Clotho | 8.8 | 9.8 | **10.1** |

(d) **Rank of LoRA**

| Dataset | 2 | 4 | 8 |
|---------|------|------|------|
| NYU-D | **65.1** | 64.4 | 64.7 |
| LLVIP | **83.9** | 78.0 | - |
| FLIR V1 | **82.9** | 74.4 | - |
| FLIR V2 | **48.0** | 45.8 | - |

(e) **Temperature for loss**

| Dataset | Learn | 0.05 | 0.1 |
|---------|-------|------|------|
| NYU-D | **65.1** | 63.0 | 62.7 |
| LLVIP | **83.9** | 81.8 | 83.1 |
| FLIR V1 | 82.9 | **83.3** | 80.3 |
| FLIR V2 | **48.0** | 45.0 | 43.2 |

(f) **Masked ratio**

| Dataset | 0.0 | 0.3 | 0.5 | 0.7 |
|---------|------|------|------|------|
| NYU-D | - | 64.8 | **65.1** | 62.7 |
| LLVIP | 80.3 | 79.9 | **83.9** | 81.5 |
| FLIR V1 | 83.5 | **84.2** | 82.9 | 81.9 |
| FLIR V2 | 43.2 | 44.0 | **48.0** | 42.5 |

## 6 CONCLUSION

In this work, we propose the LanguageBind, a language-based semantic alignment method for multimodal pretraining. We employ contrastive learning to establish modality semantic alignment between the language modality and all other modalities. To improve modal integrity, we also construct the first large-scale multi-modal dataset directly aligned to language modality, VIDAL-10M, comprising 10 million aligned VL, IL, DL, and AL pairs. Extensive experimental results, including zero-shot X-language comprehension and indirect alignment between different modalities, demonstrate the effectiveness of LanguageBind's multimodal alignment and complementary capabilities, as well as the effectiveness of VIDAL-10M.

## ACKNOWLEDGMENTS

This work was supported by the National Key R&D Program of China (2022ZD0118101), the Natural Science Foundation of China (No.62202014), Shenzhen Basic Research Program under Grant JCYJ20220813151736001, and also sponsored by CCF Tencent Open Research Fund.

## REPRODUCIBILITY STATEMENT

1. For LanguageBind approach details.
   (a) We provide a comprehensive overview of the multi-modal encoder, detailing its architecture and functionality in Section 3.1.
   (b) We outline the language encoder in Section 3.2.
   (c) We expound on the methodologies employed for multi-modal joint learning in Section 3.2

2. For VIDAL-10M dataset construction details.
   (a) We describe the procedures employed to construct the search term database in Section 4.1.
   (b) We provide insights into the strategies used for collecting and filtering video and audio data within VIDAL-10M in Section 4.2.
   (c) We elaborate on the generation of infrared and depth data, as well as the processes involved in multi-view text generation and enhancement in Section 4.3
   (d) We promise to release the VIDAL-10M dataset upon publication.

3. For setting details.
   (a) We describe in detail the training hyperparameters in Appendix B.
   (b) We describe the setup of the downstream task dataset Appendix C.

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

APPENDIX

## A  STATISTICS OF VIDAL-10M DATASET

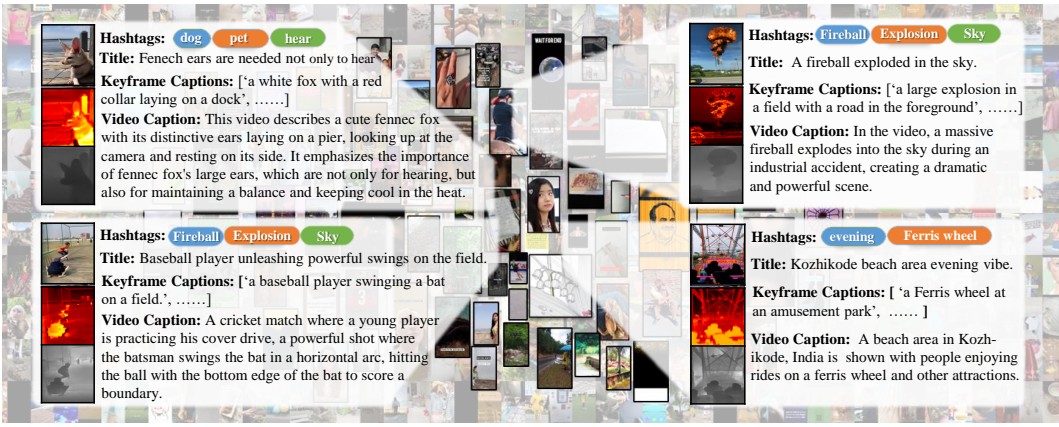

Figure 5: Examples of video-audio-text-depth-infrared pairs in VIDAL-10M, with the text components comprising hashtags, title, keyframe captions, and video caption. Examples are taken from 4 distinct clusters, corresponding to Sports, Pets & Animals, News & Politics, and Education.

In order to build a video dataset with rich visual concepts and diversity, we develop a unique but simple search term acquisition strategy. This strategy involves obtaining search terms from various visual datasets (as shown in Table 9). Subsequently, we use these search terms to gather videos from the YouTube Shorts platform, which has become a popular source for video data due to its abundance and diverse content. We collect videos in various categories, including sports, animals, nature, etc., resulting in a large and diverse dataset. Examples of video-audio-text-depth-infrared pairs in the VIDAL-10M dataset are shown in Figure 5. Moreover, to ensure data quality, we manually design a list of stop words that are filtered from our datasets. These words include terms such as "bts", "bmw", and "nfl", among others, that are not relevant to our research.

**Video categories and duration**   Furthermore, we analyze the distribution of video categories with varying durations in our datasets, as illustrated in Figure 6. The normal distribution pattern observed in this analysis indicates that our dataset covers a wide range of concepts. Besides, we show the proportions of each category across different duration grades in the VIDAL-10M dataset in Figure 7.

Table 9: Examples of textual descriptions from various datasets as search terms.

| Dataset | Search terms |
|---------|--------------|
| YouTube-8M | How to make a delicious chocolate cake.
Learn to dance salsa in 10 easy steps.
...... |
| Howto100M | How to play chess.
How to make pizza.
...... |
| ImageNet | lesser panda, red panda, panda, bear cat, cat bear, Ailurus fulgens, coon bear
killer whale, killer, grampus, sea wolf, Orcinus orca, giant panda, panda, panda bear
...... |
| COCO | A small boat floating on a body of water with a city skyline in the background.
A man with a red helmet on a small moped on a dirt road.
...... |
| Others | ...... |

Table 10: Stop words in our datasets.

| viral | funny | love | fashion | subscribe | nature |
|---|---|---|---|---|---|
| asmr | motivation | fitness | art | satisfying | foryou |
| music | india | fun | bts | amazing | edit |
| life | roblox | vlog | minecraft | design | marvel |
| explore | dubai | foryoupage | comedy | outfit | ootd |
| share | indian | lol | creative | relaxing | tattoo |
| random | instagram | quotes | workout | sad | ideas |
| views | bgmi | yummy | respect | easy | usa |
| ronaldo | jawellery | memes | happy | nfl | song |
| mlb | reel | support | nba | wow | status |
| gree | meme | gameplay | top | blackpink | whatsappstatus |
| follow | homedecor | history | tutorial | bodybuilding | japan |
| interiordesign | freefire | stunt | foodie | animation | recipe |
| skills | tips | crazy | pov | editing | aesthetic |
| style | view | london | reaction | story | pubg |
| construction | challenge | healthy | bmw | uk | free |
| hairstyle | enjoy | motivational | messi | capcut | nailart |
| entertainment | fifa | attitude | europe | health | geography |
| gta | unboxing | adventure | whatsapp | fail | btsarny |
| god | inspiration | relatable | comment | tattoos | fy |
| highlights | amazon | illustration | fortnite | ntb | avaiation |
| interior | decor | travelvlog | canada | btsarmy | tranding |
| time | mtb | luxury | vlogs | picsart | reels |
| photoshoot | business | photography | ... | ... | ... |

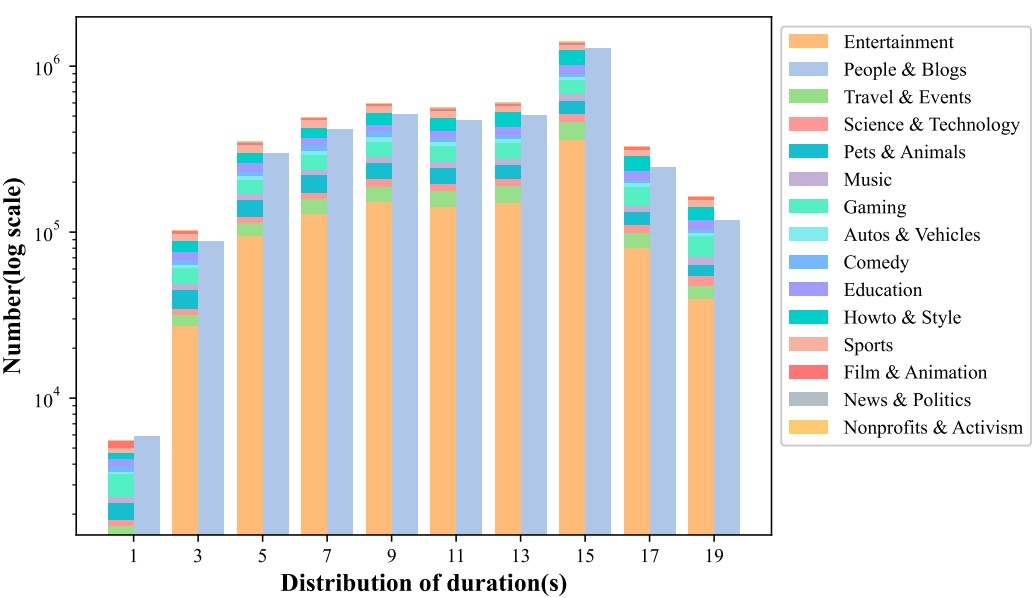

Figure 6: The number of 15 categories with different durations in our VIDAL-10M datasets. A wide range of concepts are covered.

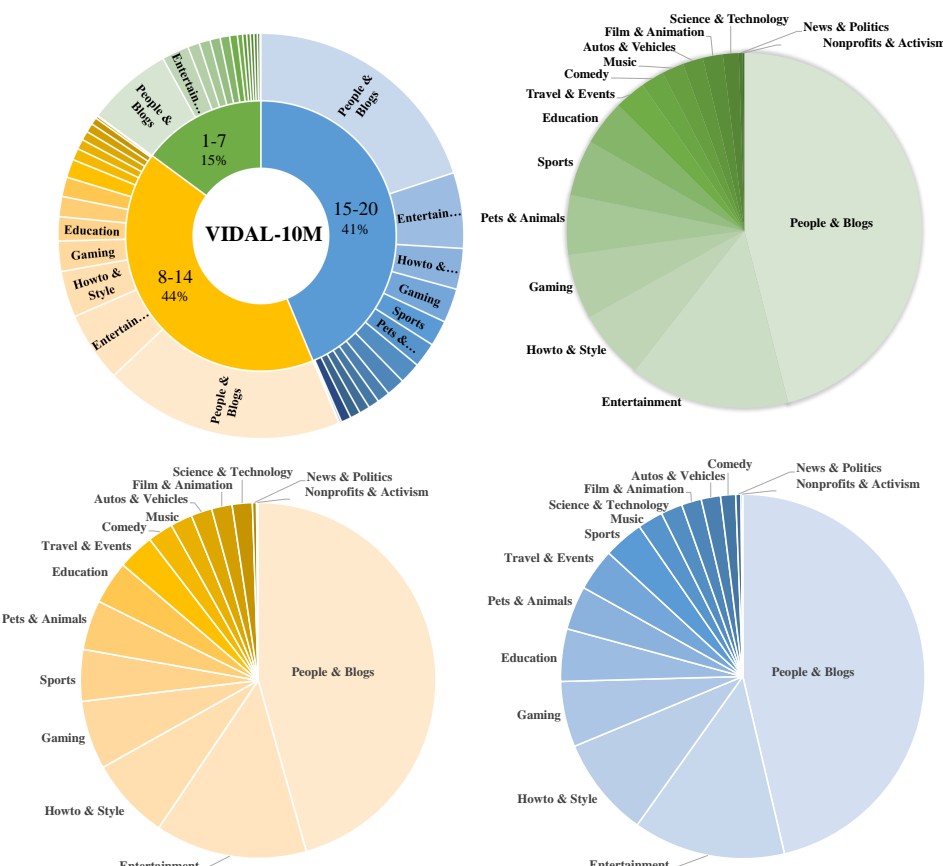

Figure 7: The statistical distribution of categories across the three duration grades in the VIDAL-10M dataset. The colors green, blue, and yellow represent video durations of 1-7, 8-14, and 15-20 s, respectively.

**FPS, Aspect ratio and Resolution**  The first aspect examined in the dataset is the Frames Per Second (FPS) of the videos. FPS refers to the number of frames or images displayed per second in a video. The aspect ratio of a video represents the proportional relationship between its width and height dimensions. It is a critical factor in determining the visual presentation and viewing experience of the videos. The distribution of FPS and aspect ratios in Figure 8 provides insights into the smoothness and fluidity of the recorded content and sheds light on the various formats and orientations used. Video resolution refers to the number of pixels in each dimension that a video contains. It directly affects the clarity, sharpness, and level of detail in the visual content. Examining the distribution of resolutions (Figure 9) in the dataset provides an understanding of the available video quality and the technological capabilities of the recorded material.

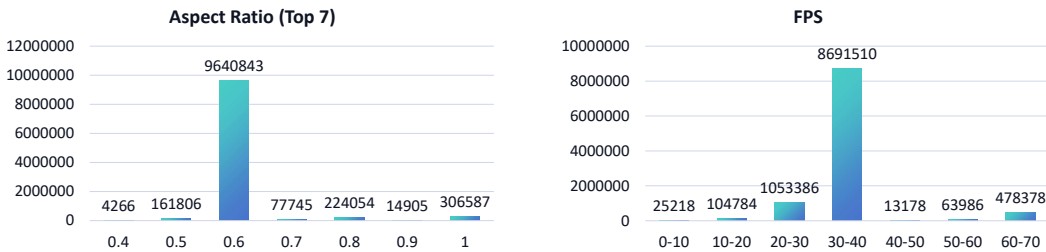

Figure 8: The distribution of FPS (Frames Per Second) and aspect ratio in the videos of the VIDAL-10M dataset.

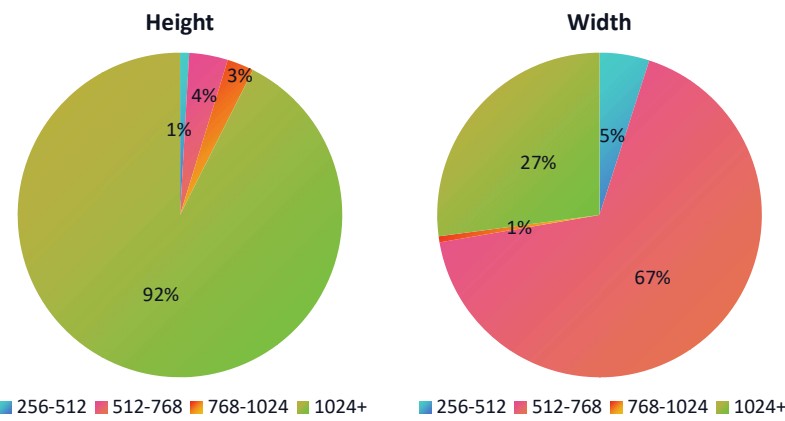

Figure 9: Height and width distribution of videos in VIDAL-10M dataset.

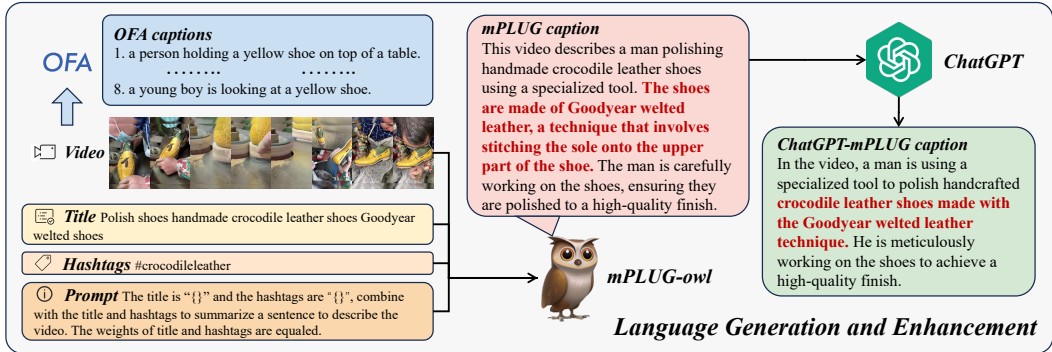

Figure 10: **Multi-view text generation and enhancement pipline**. We employ the OFA model to generate keyframe captions and input video, title and hashtags into the mPLUG-owl model to obtain video captions. The video captions are further refined using ChatGPT, resulting in the ChatGPT-mPLUG caption. The final multi-view textual description comprises these components.

# B  PRETRAINING DETAILS

In this section, we introduce our training configuration.

**Video-Language.**  For the video-text retrieval based CLIP4Clip, we verify that the *VIDAL-10M* dataset is highly aligned. We adopted the training framework of CLIP4Clip, and the model is initialized from ViT-B/32, and the rest of the parameters are the same as the default settings, except for 1 epoch and batch size of 512. For the video-text retrieval based LanguageBind, we add a temporal attention before each spatial attention following Aim (Yang et al., 2023). The temporal attention is initialized from the spatial attention and LoRA is applied only to the temporal attention. We add temporal position embedding before each temporal attention. We show the details of results as shown in Table 11. For zero-shot video classification, The text templates are sourced from OpenCLIP, with a modification consisting of the substitution of "photo" with "video" across all templates.

**Depth-Language.**  The model is initialized from OpenCLIP with a frozen language encoder. For each individual sample, we employ a random selection approach to extract either a depth image from the video sequence. Subsequently, we resize these frames to have a short edge length of 256 units, followed by a central cropping process to attain dimensions of 224×224. Additionally, we tripled the number of channels in both the depth image. The text templates employed for zero-shot classification are sourced from OpenCLIP, with a modification consisting of the substitution of "photo" with "depth photo" across all templates. This alteration yields an approximate performance gain of 1%.

Table 11: **Zero-shot Video-Text Retrieval Performance based LanguageBind.** We show the details of results.

| Dataset | Text-to-Video | | | | Video-to-Text | | | |
|---|---|---|---|---|---|---|---|---|
| | R@1↑ | R@5↑ | R@10↑ | MR↓ | R@1↑ | R@5↑ | R@10↑ | MR↓ |
| MSR-VTT | 42.6 | 65.4 | 75.5 | 2.0 | 37.9 | 63.1 | 73.3 | 3.0 |
| MSVD | 52.2 | 79.4 | 87.3 | 1.0 | 68.4 | 91.7 | 96.4 | 1.0 |
| ActivityNet | 35.1 | 63.4 | 76.6 | 3.0 | 32.3 | 62.2 | 74.5 | 3.0 |
| DiDeMo | 37.8 | 63.2 | 73.4 | 3.0 | 37.6 | 63.7 | 73.3 | 3.0 |

**Infrared-Language.** Following depth-language, it is worth noting that the text templates corresponding to infrared images retain the "photo" designation, as no discernible performance improvement is observed from this particular modification.

**Audio-Language.** The data are preprocessed as in 3.1. Unlike depth and infrared, spectrograms differ much from the domain of conventional visual images. Therefore, it is not easy to overfit during training, so we increase the training epoch and the rank of LoRA. Additionally, we replace "the/a photo of" with "the/a sound of" across all templates for audio zero-shot classification.

Table 12: Training setting.

| Config | CLIP4Clip | LanguageBind | | | |
|---|---|---|---|---|---|
| | Video | Video | Infrared | Depth | Audio |
| Vision encoder | ViT-Base/32 | ViT-Large/14 | | | |
| Optimizer | BertAdam | AdamW | | | |
| Optimizer Momentum | $\beta_1, \beta_2 = 0.9, 0.98$ | $\beta_1, \beta_2 = 0.9, 0.98$ | | | |
| Epochs | 1 | 16 | 1 | 1 | 8 |
| Learning rate | 1e-4 | 1e-4 | 1e-4 | 5e-4 | 5e-4 |
| Coefficient learning rate | 1e-3 | 1 | 1e-3 | 1e-3 | 1e-3 |
| Weight decay | 0.2 | 0.2 | | | |
| Batch size | 512 | 640 | 1024 | 1024 | 512 |
| Warmup steps | 0 | 2000 | 200 | 200 | 2000 |
| Temperature | learnable | learnable | | | |
| Learning rate schedule | cosine decay | cosine decay | | | |
| Max words | 32 | 77 | | | |
| Max frames | 12 | 8 | - | - | - |
| Mask ratio | - | 0.3 | 0.5 | 0.5 | 0.3 |
| LoRA rank | - | 16 | 2 | 2 | 16 |
| LoRA alpha | - | 16 | | | |
| LoRA dropout | - | 0.0 | 0.1 | 0.1 | 0.1 |

## C DOWNSTREAM DATASETS

**Video-language.** We perform video-text retrieval experiments on 2 datasets. **(a) MSR-VTT** (Xu et al., 2016) comprises 10K YouTube videos, each paired by 200K captions. In our analysis, we present results based on the 1K-A test subset. **(b) MSVD** (Chen & Dolan, 2011) consists of about 120K sentences and reports results on test data (670 samples).

**Infrared-language.** **(a) LLVIP** (Jia et al., 2021) constitutes a dataset for pedestrian object detection within the infrared spectrum. Following ImageBind, we extracted all people from the images, designating all other objects as background elements. This process resulted in a dataset comprising 7,622 'background' classes and 7,954 'person' classes, which was subsequently employed for binary classification testing. **(b) FLIR v1** (Teledyne FLIR, 2015a) offers comprehensive annotations for both thermal and visible spectrum frames. From the test data, we derived a dataset containing 11,696 images by extracting bounding boxes. This dataset encompasses 4 categories – ['bicycle', 'car', 'dog', 'person']. **(c) FLIR v2** (Teledyne FLIR, 2015b) includes 16,696 images after pro-

cessing similarly, which were categorized into 12 classes – ['bike', 'bus', 'car', 'hydrant', 'light', 'motor', 'other vehicle', 'person', 'sign', 'skateboard', 'stroller', 'truck'].

**Depth-language.** We use **NYU-v2 Depth-only (NYU-D)** (Silberman et al., 2012) to validate by 654 test samples. Through preprocessing, we constrained the depth images to a maximum depth of 10 meters. Following ImageBind, we undertook a category reorganization process, resulting in a total of 10 scene categories.

**Audio-language.** We validate the zero-shot classification capability with the **ESC-50** (Piczak, 2015) dataset, which has 2000 test audios, each uniquely labelled. For zero-shot retrieval, we use the **Clotho** (Font et al., 2013) dataset. Each audio has 5 corresponding captions, so we use text-to-audio retrieval to validate the model performance. We perpare test data following ImageBind.

## D  LICENSE

Unless explicitly noted otherwise, our released datasets are provided to users under the terms of the Creative Commons Attribution-NonCommercial-ShareAlike 4.0 International Public License ("CC BY-NC-SA 4.0"), in conjunction with the additional terms outlined herein. The CC BY-NC-SA 4.0 license can be accessed at `https://creativecommons.org/licenses/by-nc-sa/4.0/legalcode`. By downloading or utilizing our datasets from our website or other sources, you agree to adhere to the terms of CC BY-NC-SA 4.0, as well as the terms outlined in our dataset Terms. In the event of any conflict between the terms of CC BY-NC-SA 4.0 and our dataset Terms, the latter shall prevail. We once again emphasize that this dataset is exclusively intended for non-commercial purposes, such as academic research, teaching, or scientific publications. We strictly prohibit any commercial use of the dataset or any derived works, including the sale of data or utilization of data for commercial gain.

## E  ABLATION STUDY

In this section, we conduct extensive experiments to investigate the impact of several factors. At first, we examine the effects of different enhanced textual inputs on downstream tasks. Furthermore, we assess the impact of data volumes on pretraining. In addition, we explore various training strategies to enhance zero-shot classification. Finally, we conduct a meticulous analysis of model training configurations to ensure robust transferability.

Table 13: **Impact of different text sources.** We report the results of text-to-video R@1 for zero-shot retrieval and other datasets report top-1 accuracy. MSR-VTT results were tested on a 500K subset of VIDAL-10M. "*Raw* caption" denotes the title & hashtags.

| Modality | Dataset | *Raw* caption | *OFA* caption | *mPLUG* caption | *ChatGPT-mPLUG* caption |
|---|---|---|---|---|---|
| Video | MSR-VTT | 33.5 | 34.5 | 35.8 | **36.4** |
| Infrared | LLVIP | 83.9 | **87.2** | 84.6 | 84.8 |
| | FLIR V1 | **82.9** | 80.6 | 81.4 | 81.6 |
| | FLIR V2 | **48.0** | 45.7 | 46.8 | 46.6 |
| Depth | NYU-D | 61.5 | 62.1 | 63.9 | **65.1** |

### E.1  IMPACT OF DIFFERENT TEXT SOURCES

In Table 13, we conduct various experiments to explore how different text sources impact language modality. We verify the effectiveness of LanguageBind, trained with text from multiple sources, across various modalities. While some text sources yield good results, we discover that a single text source may not be universally suitable for all downstream tasks and datasets. In terms of video and depth modalities, the ChatGPT enhanced caption proves to be advantageous. For infrared images, the OFA performs best in the LLVIP dataset, while the raw caption achieves the highest accuracy in FLIR v1 and v2. That's why our VIDAL-10M provides multi-view textual descriptions, allowing for flexibility in selecting an appropriate text source that caters to diverse task requirements.

## E.2 SCALING THE SIZE OF DATASET

We analyze the impact of different data amounts on MSR-VTT and report the R@1 score for zero-shot retrieval as shown in Figure 11. Our findings indicate that an increase in data amount leads to significant improvement in recognition performance. Specifically, the performance of 3M ChatGPT-enhanced text surpasses that of 500k and 100k data by 0.9% and 1.6%, respectively.

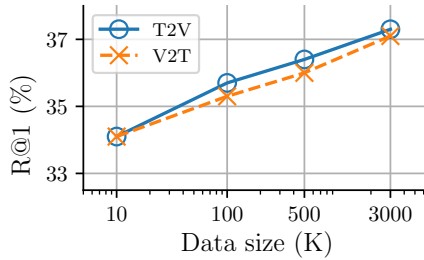

Figure 11: Scaling pretraining data size.

Furthermore, the trends observed in both video-to-text retrieval and text-to-video retrieval consistently demonstrate that the interaction between modalities plays a pivotal role in enhancing the learning process. Consequently, with the expansion of data size, the textual descriptions within the VIDAL-10M dataset align more closely with the video content and demonstrate increased scalability.

