# OpenReview forum: "LanguageBind: Extending Video-Language Pretraining to N-modality by Language-based Semantic Alignment"
_ICLR.cc/2024/Conference — ICLR 2024 poster_

### Official Review · Reviewer_ePEX · 2023-11-01

**Soundness:** 3 good
**Presentation:** 3 good
**Contribution:** 3 good
**Rating:** 6
**Confidence:** 5

**Summary:**

This paper proposes a novel pretraining paradigm called LanguageBind, which takes the language as the ind across different modalities. To this end, authors curate a large-scale multimodal dataset. Extensive experiments for different modalities demonstrate the effectiveness of the proposed paradigm.

**Strengths:**

1. The paper is clearly written and contains sufficient details and thorough descriptions of the experimental design.
2. Extensive experiments are conducted to verify the effectiveness of the proposed method and dataset.

**Weaknesses:**

1. In table 2, while authors demonstrate the improvements over ImageBind on T2V and V2T tasks, these two models are trained with different backbones, model initializations, finetuning techniques, and training data. This leads to an unfair comparison, especially considering the proposed model is leveraging more video data.

2. Based on my understanding, LanguageBind is initialized from OpenCLIP and continues to train on the VIDAL-10M dataset. Compared to OpenCLIP, it is difficult to tell whether the performance improvement comes from the proposed dataset or the new pretraining paradigm.

3. In table 4, do the authors have any intuition why raw caption works best for the Infrared modality?

**Questions:**

See the above weakness

---

> ### Author Response · Authors · 2023-11-16
>
> Thanks to your valuable comments, we have **added a number of experiments** to the paper to demonstrate the effectiveness of LanguageBind, these are shown in $\color{red}{red}$ font in revised paper.
>
>
> > **Q1**: Please provide a more fair comparison experiment in Table 2?
>
> **A1:** Thank you for your valuable feedback. We greatly appreciate your suggestion, which has allowed us to enhance the quality of our paper. In response to your recommendation, we have made several updates to the comparative model of LanguageBind in Table 2 and Table 3 in the revised version. **In Table 3, we have employed the CLIP4CLIP model structure for training on both the 380K Howto100M dataset and the 100K VIDAL-10M dataset. The results clearly demonstrate that VIDAL-10M yields superior experimental outcomes**, thereby substantiating the reliability and effectiveness of the data. Regarding Table 2, we acknowledge that the comparison between LanguageBind-Large and ImageBind-Huge may have been slightly unfair due to the novelty of the approach and the different problems in focus. To address this concern, we introduced the InternVideo model in Table 2, which used more data and a well-designed modeling structure. Our findings reveal that LanguageBind achieves remarkable performance surpassing that of InternVideo while utilizing only 1/4 of its data volume.
>
>
>
> |Table 2||||||||||||||
> |---|---|---|---|---|---|---|---|---|---|---|---|---|---|
> |Method| Dataset |MSR-VTT|||MSVD||| DiDeMo |||ActivityNet|||
> |||R@1| R@5| R@10| R@1| R@5 |R@10| R@1| R@5 |R@10| R@1 |R@5 |R@10|
> |CLIP-H/14
> |ImageBind| - |36.8| 61.8| 70.0 |-| -| -| -| - |-| - |- |-|
> |CLIP-L/14
> |UMT| 5M |33.3| 58.1| 66.7 |44.4| 73.3 |82.4 |34.0| 60.4 |68.7 |31.9 |69.2 |72.0|
> |TVTSv2 |8.5M |38.2 |62.4 |73.2 |- |-| - |34.6| 61.9 |71.5 |-| -| -|
> |InternVideo |12.8M| 40.7| -| - |43.4 |-| - |31.5 |-| - |30.7| -| -|
> |LanguageBind |3M |42.6 |65.4 |75.5 |52.2 |79.4|87.3 |37.8| 63.2 |73.4| 35.1| 63.4 |76.6|
>
>
> |Table 3||||||||
> |---|---|---|---|---|---|---|---|
> |Dataset |Method |Parameter| Source| R@1↑ |R@5↑| R@10↑ |MR↓|
> |MSR-VTT |CLIP4Clip |86M |WIT400M, HT100M-380k |32.0 |57.0 |66.9 |4.0|
> |MSR-VTT |CLIP4Clip |86M |WIT400M, VIDAL-100k |35.7| 60.8 |71.5 |3.0|
> |MSVD| CLIP4Clip| 86M| WIT400M, HT100M-380k |38.5| 66.9 |76.8 |2.0|
> |MSVD |CLIP4Clip |86M |WIT400M, VIDAL-100k| 42.0 |70.0 |79.2 |2.0|
>
>
>
>
>
>
> > **Q2**: How to verify whether the performance improvement comes from the proposed dataset or the new pretraining paradigm?
>
> **A2:** Thanks for your question. We added more ablation experiment between LanguageBind and full parameter fine-tuning and train from scratch in Table 8(c). Training from scratch exhibits the poorest performance, likely due to the lack of prior knowledge from CLIP pretraining. On multiple datasets such as LLVIP, FLIRv1, and Clotho, we found that **full-parameter fine-tuning based on CLIP weights did not perform as well as LanguageBind**. Meanwhile, the LoRA method stands out for its advantages in terms of time and memory cost. This indicates that LanguageBind is not only efficient but also effective in learning new knowledge specific to different domains.
>
> |Table 8 (c) | Training strategy|||
> |---|---|---|---|
> |  |Scratch| Full tuning| LoRA|
>  |  Time| 1.4h |1.4h |0.8h|
>  |  Mems |278M |278M |132M|
> | LLVIP |57.1| 85.1 |87.2|
>   | FLIR V1 |74.7 |81.3 |81.6|
>   | ESC-50| 86.8| 88.9| 87.4|
>   | Clotho |8.8 |9.8 |10.1|
>
>
>
>
> > **Q3**: Can the authors provide intuition why why raw caption works best for the Infrared modality?
>
> **A3:** Thanks for your question. We suppose that **for modalities with limited scene diversity, the accurate and brief text may work better, i.e., the hashtags in raw captions for depth and infrared images.** According to our research, textual data from different sources often have complementary roeeles. For modalities with diverse scenes, such as video, image, text, etc., a mixture of text from different sources may work best. While the scenerios with limited diversity shows the contrary, as the infrared modality mentioned in the quesiont. In addition, since the data of the downstream task of the infrared modality is relatively small, the fluctuation in training can easily affect the final experimental results. Thank you for your question, which is an interesting one for us to follow up and extend to other modalities we want to understand more deeply.

---

### Official Review · Reviewer_Bowz · 2023-11-06

**Soundness:** 3 good
**Presentation:** 3 good
**Contribution:** 3 good
**Rating:** 6
**Confidence:** 4

**Summary:**

The paper presents multi-modal pretraining approach with modalities N=5 (video, text, audio, depth, infrared) by using language as bind across different modalities. A frozen text encoder from a pretrained VL model is used as the feature extractor for the text modality and aligned with other modalities (pair-wise) using contrastive loss. It also introduces a dataset called VIDAL-10M with 10M data pairs from VL, DL, IL,and AL. The dataset and method is evaluated on standard retrieval benchmarks to show the effectiveness of the pretraining data as well as the technique.

**Strengths:**

1) The paper attempts to learn a unified embedding space for 5 modalities where the modalities are guided by language during pre-training. Such an embedding space can be very useful for tasks involving: i) multi-modal data for ex: video containing audio, ii) tasks where paired data is not available, for ex: Video-Infrared, Video-depth etc

2) The paper introduces a dataset with 10M paired data from AL, VL, IL and DL which is important for driving research in the multimodal learning area as many of the real-world applications contain multimodal data. It follows a careful approach by leveraging existing vision and language models (OFA, mPLUG-owl, chatgpt etc) to collect a balanced (in topic) and diverse (in semantic) data-pairs.

3) The introduced dataset and the pretrained model is shown to be useful for:
a) cross-modality video retrieval task where it outperforms its counterparts (ImageBind, CLIP-straight, CLIP4clip).
b) AL, DL, IL zero-shot classification tasks.
This shows that the model has learned good representations in the joint embedding space.

**Weaknesses:**

1) It is not clear from the text or Table2, the size of the pretraining data used for MSR-VTT and MSVD datasets. For a fair comparison, all methods should be pretrained with same amount of data but here CLIP-Straight is trained with WIT400M only (initialized from CLIP but no fine-tuning), CLIP4clip is trained with WIT400M+HT100M-380k whereas the proposed technique (although CLIP4clip technique is used) is pretrained with WIT400M+VIDAL-10M. It would be a fair comparison if all methods use similar sized data, i.e what would be the performance of other technique like CLIP4clip if additional data (not VIDAL-10M) is used for training.

2) One of the goals of learning a model from multimodal data is  that the data can use all available modalities to learn stronger representations but there are no experiments to demonstrate this, for ex: instead of using just video -> text retrieval, it would interesting to show that video+audio -> text retrieval has better performance.

3) There are few other advantages of multimodal learning in situations where:
a) one of the modalities is corrupted
b) one of the modalities has some weaknesses (videos taken in the dark, OR audio from multiple sources)
c) one of the modality undergoes a domain change while the other doesn't (eg: videos under weather changes etc)
but none of these has been addressed in this paper. It would be interesting to see results on at least one of the above scenarios.

4) It would also be interesting to see an experiment where the model is evaluated on retrieval task where the modalities doesn't contain text. For ex (video<->audio, video<->infrared). This will evaluate the quality of learned representations.

**Questions:**

I would like authors to discuss all the points described above.

Final Rating: After reading the rebuttal and comments from other reviewers, I have decided not to change the score.

---

> ### Author Response · Authors · 2023-11-16
>
> Thanks to your valuable comments, we have **added a number of experiments** to the paper to demonstrate the effectiveness of LanguageBind, which are shown in $\color{red}{red}$ font in revised paper.
>
>
> > **Q1**: Please clarify the fairness of the comparison experiments in Table 2?
>
> **A1:** Thank you for your valuable feedback. We greatly appreciate your suggestion, which has allowed us to enhance the quality of our paper. In response to your recommendation, we have made several updates to the comparative model of LanguageBind in Table 3 in the revised version. **In Table 3, we have employed the CLIP4CLIP model structure for training on both the 380K Howto100M dataset and the 100K VIDAL-10M dataset. The results clearly demonstrate that VIDAL-10M yields superior experimental outcomes**, thereby substantiating the reliability and effectiveness of the data. More experimental results on data scaling based on CLIP4Clip are presented in Appendix E.2.
>
> Table 3||||||||
> |---|---|---|---|---|---|---|---|
> |Dataset |Method |Parameter| Source| R@1↑ |R@5↑| R@10↑ |MR↓|
> |MSR-VTT |CLIP4Clip |86M |WIT400M, HT100M-380k |32.0 |57.0 |66.9 |4.0|
> |MSR-VTT |CLIP4Clip |86M |WIT400M, VIDAL-100k |35.7| 60.8 |71.5 |3.0|
> |MSVD| CLIP4Clip| 86M| WIT400M, HT100M-380k |38.5| 66.9 |76.8 |2.0|
> |MSVD |CLIP4Clip |86M |WIT400M, VIDAL-100k| 42.0 |70.0 |79.2 |2.0|
>
>
>
> > **Q2**: How to validate that the data can use all available modalities to learn stronger representations?
>
> **A2:** This is an intriguing question. In Table 6, we guided joint retrieval experiments and demonstrated that the results of **Video+Audio->Text, Depth+RGB->Text, and Infrared+RGB->Text outperform the performance of individual modalities.** These results validate that LanguageBind has learned a unified semantic space and can utilize all available modalities to acquire stronger representations.
>
> |Table 6||||
> |---|---|---|---|
> |Dataset| Method| Task |Top-1|
> | MSRVTT| ImageBind |   V→T |36.1∗|
> | MSRVTT| ImageBind |   A+V→T | 36.8 (+0.7)|
> | MSRVTT| Ours|   V→T |41.4|
> | MSRVTT| Ours|   A+V→T |42.0 (+0.6)|
> | NYU| ImageBind |   D→T |54.0|
> | NYU| Ours|   D→T |65.1|
> | NYU| Ours|  RGB→T |76.0|
> | NYU| Ours|   D+RGB→T| 77.4 (+1.4)|
> | LLVIP| Ours|   RGB†→T |62.4|
> | LLVIP| Ours|   I+RGB†→T |79.3 (+16.9)|
>
>
>
> > **Q3**: Can LanguageBind address one of the following problems: a) one of the modalities is corrupted; b) one of the modalities has some weaknesses; c) one of the modality undergoes a domain change while the other doesn't?
>
> **A3:** LanguageBind learns a unified representation using available modalities, which enables the model to exhibit strong robustness when a modality undergoes a domain change. For instance, in the downstream task of nighttime pedestrian detection sourced from the LLVIP dataset, **RGB images captured in the dark are used.** In Table 6, we demonstrate that the performance of dark images->text is 62.4. When infrared is incorporated, the performance increases to 79.3. This showcases the advantage of binding multiple modalities into a semantic space, as it demonstrates robustness when another modality undergoes changes a domain change.
>
>
> > **Q4**: Can authors provide experiments for retrieval tasks where the modalities doesn't contain text?
>
> **A4:** Yes, to evaluate the quality of the learned unified representation, **we present the results of emergent zero-shot retrieval in Table 7. This includes Video -> Audio, Image -> Infrared, Image -> Depth, and the performance when incorporating text embedding.** On each benchmark, the performance of emergency zero-shot retrieval achieves significant gains, even approaching results obtained by incorporating textual features. These results suggest that LanguageBind aligns various modalities and implicitly transfers text supervision associated with specific modalities and tasks.
>
>
>
> |Table 7|||||
> |---|---|---|---|---|
> |Dataset| Method| Task |Emergent|R@1|
> |AVE† |Ours| V→A |✔ |10.6|
> |AVE†|ImageBind| V→A  |✗ |36.9|
> |VGGS†|Ours| V→A  |✔ |10.0|
> |VGGS†|ImageBind| V→A  |✗ |28.7|
> |LLVIP†|Ours| RGB→I |✔| 7.5|
> |LLVIP†|Ours| RGB+T→I |✗ |9.1|
> |LLVIP†|Ours| I→RGB |✔| 9.3|
> |LLVIP†|Ours| D+I→RGB |✗ |10.6|
> |NYU|Ours| RGB→D |✔| 17.9|
> |NYU|Ours| RGB+T→D |✗ |18.3|
> |NYU|Ours| D→RGB|✔ |24.5|
> |NYU|Ours| D+T→RGB| ✗ |25.7|

---

### Official Review · Reviewer_ZKDg · 2023-11-08

**Soundness:** 3 good
**Presentation:** 3 good
**Contribution:** 2 fair
**Rating:** 6
**Confidence:** 4

**Summary:**

The paper proposes LanguageBind, a method for training encoders of multiple modalities in a joint embedding space by aligning them to a frozen text encoder. Additionally, the authors introduce VIDAL-10M, a multimodal dataset that includes data for 4 modalities with paired textual descriptions. The authors rely on multiple third-party tools for creating VIDAL like OFA, mPLUG-Owl, and ChatGPT, in addition to modality-specific generation modules to collect data for the infrared and depth modalities. Various techniques are utilized to train LanguageBind like LoRA tuning, masking, and initializing from pre-trained CLIP checkpoints. The authors provide zero-shot retrieval and recognition experiments to showcase the effectiveness of their method.

**Strengths:**

1. The VIDAL dataset is potentially interesting. In particular the utilization of multiple captioning models to enhance the textual descriptions for spatial and temporal information (OFA and mPLUG) as well as ChatGPT for refining the descriptions. This is further confirmed by the results of the video modality in Table.4
2. Overall, the results reported by the authors for the different modalities+text benchmarks are strong and reflect a good performance of the model.

**Weaknesses:**

1. The authors try to draw parallels to ImageBind (method name, comparisons, frequent mentions in the abstract and throughout the paper). However, LanguageBind is much closer to standard CLIP training where a joint encoder is trained between textual descriptions and sensory data for a certain modality. There have been various examples of such methods ever since CLIP was introduced for image-text pre-training such as AudioCLIP, PointCLIP, VideoCLIP (only to mention a few). This is very important because the paper only includes evaluations testing the performance of each modality and text which is different than ImageBind's proposal of testing alignment that emerged indirectly by training the modalities jointly. For LanguageBind, the benefit of training all modalities in a joint embedding space is not showcased.
2. The technical contributions of the paper are weak in terms of novelty and potential interest to the wider research community. The method is more a bag of well-established tricks (e.g. masking from FLIP, LoRA tuning, fine-tuning openCLIP checkpoints)
3. While the VIDAL dataset is potentially interesting, the fact that all modalities other than Video and Audio are automatically generated by off-the-shelf models is concerning in terms of its long-term impact. (the dataset will likely be outdated once higher fidelity generation models are developed).

**Questions:**

- The paper only includes LoRA results. Why are not there any full fine-tuning results given the authors collected a decent-sized dataset (10M across modalities) ?
- Similar to the previous point, what happens if the modality encoders are not initialized with openCLIP checkpoints similar to ImageBind where only the text and image encoders are pre-trained ?
- In the ablation, the performance drops when moving from 0.5 -> 0.3 masking which is counter-intuitive. What is the authors' explanation? What happens with no masking is used?
- As stated above, what is the value of training all modalities in a joint embedding space if all use cases have to do with text only?

---

> ### Author Response · Authors · 2023-11-16
>
> We greatly appreciate your valuable feedback, which has led us to include some additional meaningful and interesting experiments in our paper. You can find them highlighted in $\color{red}{red}$ font in the revised version of the paper. We would greatly appreciate it if you could take the time to review the revised document.
>
> > **Q1**: What is the benefit of training all modalities in a joint embedding space and the difference of previous CLIP-based works?
>
> **A1:** In order to effectively showcase the indirect alignment and multi-modal complementary capabilities of LanguageBind, we have added relevant experimental findings, as depicted in Table 6 and Table 7.
>
> - In Table 6, we guided joint retrieval experiments and demonstrated that the results of **Video+Audio->Text, Depth+RGB->Text, and Infrared+RGB->Text outperform the performance of individual modalities.**
>
>
> |Table 6||||
> |---|---|---|---|
> |Dataset| Method| Task |Top-1|
> | MSRVTT| ImageBind |   V→T |36.1∗|
> | MSRVTT| ImageBind |   A+V→T | 36.8 (+0.7)|
> | MSRVTT| Ours|   V→T |41.4|
> | MSRVTT| Ours|   A+V→T |42.0 (+0.6)|
> | NYU| ImageBind |   D→T |54.0|
> | NYU| Ours|   D→T |65.1|
> | NYU| Ours|  RGB→T |76.0|
> | NYU| Ours|   D+RGB→T| 77.4 (+1.4)|
> | LLVIP| Ours|   RGB†→T |62.4|
> | LLVIP| Ours|   I+RGB†→T |79.3 (+16.9)|
>
>
> - **In table 7, we present the results of emergent zero-shot retrieval in Table 7. This includes Video -> Audio, Image -> Infrared, Image -> Depth, and their performance when incorporating text embedding.** On each benchmark, the performance of emergent zero-shot retrieval achieves significant gains, even approaching results obtained by incorporating textual features.
>
>
>
> |Table 7|||||
> |---|---|---|---|---|
> |Dataset| Method| Task |Emergent|R@1|
> |AVE† |Ours| V→A |✔ |10.6|
> |AVE†|ImageBind| V→A  |✗ |36.9|
> |VGGS†|Ours| V→A  |✔ |10.0|
> |VGGS†|ImageBind| V→A  |✗ |28.7|
> |LLVIP†|Ours| RGB→I |✔| 7.5|
> |LLVIP†|Ours| RGB+T→I |✗ |9.1|
> |LLVIP†|Ours| I→RGB |✔| 9.3|
> |LLVIP†|Ours| D+I→RGB |✗ |10.6|
> |NYU|Ours| RGB→D |✔| 17.9|
> |NYU|Ours| RGB+T→D |✗ |18.3|
> |NYU|Ours| D→RGB|✔ |24.5|
> |NYU|Ours| D+T→RGB| ✗ |25.7|
>
>
> - These experimental results affirm that LanguageBind not only exhibits commendable indirect alignment proficiency but also demonstrates remarkable modal complementarity. This substantiates the exceptional quality of LanguageBind's alignment within the shared semantic space.
>
>
> > **Q2**: What is the main contribution of this work?
>
> **A2:** Our primary objective behind proposing LanguageBind is to establish alignment between various modalities and a Language-centric semantic space. This approach is motivated by the fact that numerous contemporary multimodal downstream tasks are inherently interconnected with Language. **In table 6 and table 7, by aligning modalities within a shared semantic space, we can effectively leverage the complementarity and alignment between these modalities**, thereby enhancing the performance of individual modalities in downstream tasks, as discussed in the initial question. Furthermore, aligning other modalities to the Language modality offers a potential advantage:  LanguageBind has the potential to facilitate the development of LLM-driven full-modal Large Models, following the footsteps of PandaGPT, LLaMa-AdapterV1, NeXtGPT, and similar research initiatives.
>
>
>
> > **Q3**: What is the significance and future development of proposed VIDAL dataset, since part of the modalities are automatcally generated?
>
> **A3:** When it comes to building depth and infrared images using generative models, there are several key considerations that we have taken into account. These considerations, among others, include the following two points:
>
> - Limited data size and categorical labels: Depth and infrared data often exhibit a relatively small-scale dataset with text labels primarily consisting of categorical words. This characteristic may not be ideal for large-scale pre-training.
>
> - Evolving generative models for data generation: The field of generative models is constantly evolving, and **we have observed an increasing number of works utilizing generative models for data creation.** For instance, BLIP2 employs the BLIP-large captioning model to generate image captions. While the depth and infrared data in VIDAL-10M seems have a long-term impact, the slow progress in both data development and the iteration of generative models in these domains suggests that the relevant data in VIDAL-10M will continue to make significant contributions in the long term. Additionally, we can draw inspiration from the data mixing method employed by BLIP2, where the depth, infrared, and other modality data in **VIDAL-10M can be combined with real scene data.** This mixing approach has the potential to effectively leverage the strengths of different modalities and enhance their usefulness.

---

> ### Author Response · Authors · 2023-11-16
>
> > **Q4**: How the training strategy influence the training results? To be specific, what is the results of full tuning? And what happens if the modality encoders are not initialized with openCLIP checkpoints
>
> **A4:** Considering that our VIDAL-10M dataset is relatively large, we finally adopted the LoRA technique and the Vit-Large framework instead of Vit-Huge like ImageBind in order to accelerate the training speed and save computational resources. In view of your helpful suggestions, we decide to conduct some full tuning experiments to demonstrate the experimental performance of our method in this aspect. **As indicated by Table 8, we compare three different initialization strategies.** Training from scratch exhibits the poorest performance, likely due to the lack of prior knowledge from CLIP pretraining. On the other hand, full tuning shows significant improvement compared to training from scratch. This highlights the positive impact of leveraging prior knowledge in the form of pre-trained weights. Meanwhile, the LoRA method stands out for its advantages in terms of time and memory cost. It requires less time and memory resources compared to full tuning. Furthermore, LoRA outperforms full tuning on multiple datasets such as LLVIP, FLIRv1, and Clotho. **This indicates that LoRA is not only efficient but also effective in learning new knowledge specific to different domains** while better retaining the previously acquired knowledge from the pre-trained OpenCLIP models.
>
> |Table 8| Training strategy |||
> |---|---|---|---|
> |  |Scratch| Full tuning| LoRA|
>  |  Time| 1.4h |1.4h |0.8h|
>  |  Mems |278M |278M |132M|
> | LLVIP |57.1| 85.1 |87.2|
>   | FLIR V1 |74.7 |81.3 |81.6|
>   | ESC-50| 86.8| 88.9| 87.4|
>   | Clotho |8.8 |9.8 |10.1|
>
>
> > **Q5**: Why the preformance drops when moving from 0.5 -> 0.3 masking? What happens witho no masking?
>
> **A5:** In our ablation experiments, we conducted experiments with a mask ratio of 0 and validated them on additional datasets. Interestingly, it is abnormal that performance decreases with an increase in the visible area for human understanding of images. However, from a model learning perspective, **particularly in terms of data augmentation and regularization**, applying a certain ratio of masking to images is a natural form of data augmentation. It forces the model to extract features from different patch tokens with each instance. **A similar example is FLIP, where they achieved better results in image-text retrieval with a mask ratio of 0.5 compared to a mask ratio of 0.0**, which aligns with our findings.

---

> ### Author Response · Authors · 2023-11-18
>
> > **Q6**: what is the value of training all modalities in a joint embedding space if all use cases have to do with text only?
>
> **A6:** We have taken your advice into consideration and made significant improvements to our study. Specifically, **we have added indirect alignment and multi-modal complementary experiments in Table 6 and Table 7** to further validate the value of our research. Aligning modalities within a shared language semantic space not only benefits various language-based downstream tasks but also enables us to harness the complementarity and alignment between modalities, resulting in improved performance of individual modalities, as demonstrated in Table 6 and Table 7. For instance, in the downstream task of nighttime pedestrian detection sourced from the LLVIP dataset, **RGB images captured in the dark are used.** In Table 6, we demonstrate that the performance of dark images->text is 62.4. When infrared is incorporated, the performance increases to 79.3. This showcases the advantage of binding multiple modalities into a semantic space, as it demonstrates robustness when another modality undergoes changes a domain change.
>
>
>
> |Table 6||||
> |---|---|---|---|
> |Dataset| Method| Task |Top-1|
> | MSRVTT| ImageBind |   V→T |36.1∗|
> | MSRVTT| ImageBind |   A+V→T | 36.8 (+0.7)|
> | MSRVTT| Ours|   V→T |41.4|
> | MSRVTT| Ours|   A+V→T |42.0 (+0.6)|
> | NYU| ImageBind |   D→T |54.0|
> | NYU| Ours|   D→T |65.1|
> | NYU| Ours|  RGB→T |76.0|
> | NYU| Ours|   D+RGB→T| 77.4 (+1.4)|
> | LLVIP| Ours|   RGB†→T |62.4|
> | LLVIP| Ours|   I+RGB†→T |79.3 (+16.9)|
>
>
> |Table 7|||||
> |---|---|---|---|---|
> |Dataset| Method| Task |Emergent|R@1|
> |AVE† |Ours| V→A |✔ |10.6|
> |AVE†|ImageBind| V→A  |✗ |36.9|
> |VGGS†|Ours| V→A  |✔ |10.0|
> |VGGS†|ImageBind| V→A  |✗ |28.7|
> |LLVIP†|Ours| RGB→I |✔| 7.5|
> |LLVIP†|Ours| RGB+T→I |✗ |9.1|
> |LLVIP†|Ours| I→RGB |✔| 9.3|
> |LLVIP†|Ours| D+I→RGB |✗ |10.6|
> |NYU|Ours| RGB→D |✔| 17.9|
> |NYU|Ours| RGB+T→D |✗ |18.3|
> |NYU|Ours| D→RGB|✔ |24.5|
> |NYU|Ours| D+T→RGB| ✗ |25.7|
>
>
>
> Furthermore, LanguageBind offers additional potential value by facilitating better alignment of other modality encoders with language. This alignment creates a more convenient framework for subsequent researchers to utilize our modal encoder in their explorations of Large Language Models and related research endeavors. We encourage other researchers to delve into similar areas of study or leverage LanguageBind's modal encoder in combination with LLM to develop cross-modal large models, such as NextGPT, PandaGPT, etc., which represents a primary focus of our ongoing research.

---

> > ### Comment · Reviewer_ZKDg · 2023-11-21
> >
> > I would like to thank the authors for their efforts in the rebuttal. The authors have significantly improved the paper based on the reviewers' feedback and addressed many of my concerns. In particular, they have showcased the benefits of training in a joint embedding space (Tables 6 and 7). Additionally, they have expanded the ablation study to include design choices like training from scratch, full-finetuning, and no-masking. While the paper has improved during the discussion period, I still believe the contributions provided in the paper have limited novelty. Therefore, I will raise my score to reflect the improvements in the most updated draft.

---

### Official Review · Reviewer_fQRK · 2023-11-09

**Soundness:** 3 good
**Presentation:** 2 fair
**Contribution:** 3 good
**Rating:** 8
**Confidence:** 3

**Summary:**

The authors introduce LanguageBind, an alternative approach to ImageBind where language is the primary modality that all other modalities are aligned to (instead of images). They also introduce a new dataset called VIDAL-10M that contains language-aligned data for visual, infrared, depth, and audio modalities. LanguageBind achieves state-of-the-art zero-shot classification on various infrared, depth, and audio benchmarks, as well as zero-shot video-text retrieval. Finally, they analyze the impact of scaling their dataset size on MSR-VTT R@1 and provide some training ablations that measure changes in zero-shot classification on NYU-D as a function of training epochs, batch size, LoRA rank, loss temperature, and masking ratio.

**Strengths:**

* While ImageBind was able to perform zero-shot classification by pairing its text encoder with other modalities, even if they hadn't been observed together during training, LanguageBind takes an alternative approach by obtaining the text-aligned data via synthesizing the rarer modalities from visual information in their collected text-paired data, and then training the model to align each modality separately to text. This is a subtle but interesting distinction.
* The authors describe their data collection pipeline and state they will release upon publication, which would be valuable for the broader community.
* The authors overcome the lack of pair infrared/depth data with other modalities by utilizing pretrained generative models for synthesizing a large-scale dataset is an interesting research direction that is currently gaining popularity. The VIDAL-10M dataset could be a fruitful playground for future research on scaling synthetic data generation.
* The scaling curves presented in Figure 5 are promising. They suggest one could continue scaling the techniques in this paper to continue advancing the state-of-the-art.
* Typically, models like CLIP are trained with the goal of producing a strong image encoder. It is interesting that LanguageBind is able to achieve competitive results by aligning to the text encoder instead.

**Weaknesses:**

* __Lack of ablations__: The authors only provide a limited set of ablations for a single modality (depth) on a single dataset (NYU-D). It is not clear whether the ablated decisions would impact other datasets or modalities. This is especially true because of the results in Table 4, which suggest each modality and dataset responds differently to the kinds of text annotations used, as stated by the authors. Providing a small amount of ablations on just one of these combinations makes the paper seem incomplete. Furthermore, since this paper is explicitly comparing to ImageBind, which provides extensive ablations, this paper would be much more convincing with a broader set of ablations to match.
* __Model release__: it is unclear whether the authors intend to release their models, which is a bit unexpected since they state they will release the dataset, and the model weights should be fairly small since they are mostly LoRA modules applied to OpenCLIP.

**Update**: the authors have incorporated more ablations in the rebuttal, along with a statement on model release, that adequately address my concerns. I have increased my score on "soundness" to "good" and my overall rating to "accept, good paper" to reflect this.

**Questions:**

### Audio Processing

I don't understand how the authors are processing the audio.

They state "For example, a 4-second spectrogram would be repeated twice and then padded with zero for an additional 2 seconds." Why isn't a 4-second spectrogram simply padded with zero for the remaining 6 seconds? ImageBind does not repeat spectrograms. From ImageBind Appendix B.1: "For audio, we process each raw audio waveform by sampling it at 16KHz followed by extracting a log mel spectrogram with 128 frequency bins using a 25ms Ham- ming window with hop length of 10ms. Hence, for a t second audio we get a 128 ×100t dimensional input."

I'm also confused about this sentence: "If the duration exceeds 10 seconds, we randomly sample three 10-second audio segments, each from the front 1/3, middle 1/3, and back 1/3 of the original audio, and finally stack them together.". What is being stacked along what dimension exactly?

### VIDAL-10M

* How are the multi-view text annotations used during training? Randomly sampled at each step? There's also ambiguous wording later on like in section 6.1: "allowing for flexibility in selecting an appropriate text source that caters to diverse task requirements." How are the authors selecting "an appropriate text source" during training?
* There already exist short-video datasets like WebVid-10M, as the authors mention. Why not just use the sRGB-TIR/GLPN models, as well as OFA/mPLUG-Owl on those existing datasets instead of constructing this new one? At first, I thought the motivation for VIDAL-10M was to obtain a multimodal dataset with more modalities than existing datasets, but the "new" modalities (infrared, depth) are just generated by these models. Not clear why you need to collect the audio/video in the first place if that's the case. WebVid videos have an average duration of 18 seconds, which seems similar to VIDAL-10M. Perhaps I'm missing some of the details here, but if so it might be beneficial to highlight these differences more clearly.

### Miscellaneous

* Section 3.2: this does not say what pooling method is used to go from text logics of length L to single normalized text vector. Typically this is done with either CLS token pooling, max pooling, or mean pooling, but the authors do not mention that here.
* Table 3: ImageBind uses OpenCLIP. How are their numbers on LLVIP (63.4) worse than the reported OpenCLIP numbers in this table (82.2)?

---

> ### Author Response · Authors · 2023-11-16
>
> We greatly appreciate your valuable feedback. As a result, we have incorporated several additional experiments into the paper to showcase the effectiveness of LanguageBind. These newly added experiments are highlighted in $\color{red}{red}$ font within the revised version of the paper.
>
> ## Weakness
> > **Weakness 1**:  Lack of ablations
>
> **Answer:** We have added more ablation experiments, including more modalities, more datasets, and other aspects, which are highlighted in red in the paper.
>
> |  Training epochs | | |Training batch size | |||Rank of LoRA | |||
> |---|---|---|---|---|---|---|---|---|---|---|
> |  Dataset |  1 | 5  |  Dataset |512| 1k |2k |  Dataset |2| 4| 8|
> |  NYU-D   |  65.1 |  64.5 |  NYU-D |63.9 |65.1 |64.5 |   NYU-D |65.1| 64.4 |64.7|
> |  LLVIP |83.9 |81.1 |  LLVIP |80.0 |83.9 |78.6 |  LLVIP |83.9 |78.0 |- |
> |  FLIR V1|82.9 |85.0 | FLIR V1 |81.6 |82.9| 85.2 |   FLIR V1 |82.9| 74.4 |- |
> |  FLIR V2| 48.0| 44.7  |  FLIR V2| 45.1| 48.0 |47.9  |  FLIR V2| 48.0| 45.8| -  |
>
> |Temperature for loss | |||Masked ratio | ||||
> |---|---|---|---|---|---|---|---|---|
> |  Dataset |Learn |0.05 |0.1 |  Dataset |0.0 |0.3 |0.5| 0.7|
>  |  NYU-D |65.1 |63.0 |62.7 |   NYU-D |- |64.8| 65.1| 62.7|
>  |  LLVIP |83.9| 81.8 |83.1 |  LLVIP |80.3| 79.9| 83.9| 81.5|
> | FLIR V1 |82.9| 83.3| 80.3 |   FLIR V1 |83.5 |84.2| 82.9| 81.9|
>   |  FLIR V2| 48.0 |45.0|43.2  |  FLIR V2| 43.2 |44.0 |48.0 |42.5|
>
> |Training strategy | |||
> |---|---|---|---|
> |  |Scratch| Full tuning| LoRA|
>  |  Time| 1.4h |1.4h |0.8h|
>  |  Mems |278M |278M |132M|
> | LLVIP |57.1| 85.1 |87.2|
>   | FLIR V1 |74.7 |81.3 |81.6|
>   | ESC-50| 86.8| 88.9| 87.4|
>   | Clotho |8.8 |9.8 |10.1|
>
> > **Weakness 2**:  Model release
>
> **Answer:**
> We have open-sourced the LanguageBind pre-training model, but due to the rules of this conference, I can't provide you with the address information at this time, thank you for your understanding！
>
> ## Questions
> > **Q1**: How the audio is processed?
>
> **A1:** Sorry for the confusion, we're here for further clarification.
>
> - **The motivation of audio processing.** ImageBind is trained only on AudioSet, where all the audio clips are 10 seconds long. Therefore, padding does not have a significant impact for them. However, our dataset consists of various durations. Hence, we first perform repetition. If the remaining duration is not sufficient for repetition, it will be padded with zeros. This approach largely **protects the model's learning of audio features from a stable data domain** and efficiently utilizes computational resources. Additionally, for audio clips longer than 10 seconds, while ImageBind does not encounter this issue, we aim to **preserve the original information of the audio**. Therefore, we randomly sample three audio segments. To leverage the pretraining weights of the model, which are trained on RGB images, this is why we use three segments instead of any other number.
>
> - **Audio stacking.** We apologize for any confusion caused and please allow us to clarify in detail. Now we obtain three audio segments, and these segments are first converted into spectrograms. Assuming each spectrogram has a shape of (num_mel, target_len), we stack the three spectrograms together to obtain the final input with a shape of (3, num_mel, target_len).
>
>
> > **Q2**: How are the multi-view text annotations used during training?
>
> **A2:** **Multi-view text annotations are randomly sampled at each step.** In our abalation experiments, we found that the source of the best performing text data is inconsistent for different downstream tasks in different modalities, e.g., infrared, depth. The OFA caption text performs best on the LLVIP dataset for infrared, the raw caption text performs best on the FLIR dataset for infrared, and the ChatGPT refined text performs best on the NYU dataset for depth and the MSR-VTT dataset for video. This suggests that text from different sources has its own unique strengths, so we end up providing text data with multiple perspectives in our VIDAL-10M dataset. According to our training experience, for modality data such as video with a wide range of scenes, mixing multi-view texts for training gives the best results, while for modal data such as depth and infrared thermal images with limited scenes, there may be uncertainty in the source of the best performing text, but the performance of the mixed multi-view text is also very good. More details can be found in our ablation experiments, where we have added more detail.

---

> ### Author Response · Authors · 2023-11-18
>
> > **Q3**: Why collect a new VIDAL rather than use those existing datasets?
>
> **A3:** we have taken this into consideration when building the dataset. The conditions we considered when selecting the video data include, but are not limited to, the following 1.each video is without a watermark, ideally. 2. show the whole event within 20s, preferably without truncation 3. high video resolution 4. visual diversity. The Webvid-10M dataset is  an excellent work, **but since every video in this dataset has the "Shutterstock" watermark, it severely affects the generated depth map and infrared.** Therefore, the current video dataset does not meet the above conditions, and we reconstruct the video part of VIDAL-10M.  We crawl videos from short video platforms, which have full semantics and have been filtered by users, rather than truncated segments from longer videos.
>
> > **Q4**: What method is used to go from text logics of length L to single normalized text vector?
>
> **A4:**  Thanks for reminding, and we have used the [CLS] token here.
>
> > **Q5**: Why is OpenLIP in ImageBind worse than reported OpenCLIP in LLVIP?
>
> **A5:** Since imagebind uses images as intermediates to align infrared and language, **the alignment of thermal to language is indirect,** whereas OpenCLIP directly aligns the visual to the language, so even though there is no alignment operation on the thermal data, the advantage of direct alignment to the language semantics instead allows OpenCLIP to outperform ImageBind on the LLVIP dataset. What's more, this is one reason why we constructed the VIDAL-10M infrared-text pairs, as we feel that the current multi-modal downstream task is closely related to language, and that a language-driven shared semantic space may be more suitable for the current multimodal development.

---

### Author Response · Authors · 2023-11-18
**General Response to ACs**

We sincerely thank the reviewers for their detailed and valuable feedback. All reviewers (fQRK, ZKDg, Bowz, ePEX) appreciate the LanguageBind model's outstanding performance on extensive multimodal tasks and the contribution of the VIDAL-10M dataset to the community's future development. Some reviewers also appreciate the interestingness of the VIDAL-10M dataset (fQRK, ZKDg) and the LanguageBind model (fQRK). Furthermore, we have added some experiments in the latest version of the paper, demonstrating the superior performance of the LanguageBind model in aligning multiple modalities indirectly and facilitating collaboration and complementarity among different modalities.

Based on those comments, we've added some noteworthy replies for the reviewers including.

- **[Reviewer fQRK]** We added Table 8, which presents a comprehensive set of ablation experiments.
- **[Reviewer fQRK]** We clarified the audio preprocessing, provided the motivation for collecting VIDAL-10M, discussed the details of model training, and compared it with ImageBind.

- **[Reviewer ZKDg]** We clarified the benefits of aligning to the language space, such as facilitating the integration of additional modalities into large language models.
- **[Reviewer ZKDg]** We added Table 3 to clarify the effectiveness of the dataset. Then, we clarified the applications of the generated dataset in various aspects.
- **[Reviewer ZKDg]** We added Table 8 (c) to discuss the impact of different training strategies on the results. We compared the results of training from scratch without loading OpenCLIP, full tuning, and LoRA tuning, demonstrating the effectiveness of LoRA tuning.
- **[Reviewer ZKDg]** We discussed the impact of the mask ratio on the model.

- **[Reviewer ePEX]** We added Table 8 (c) to provide a comparison between training from scratch, full tuning, and LoRA tuning, demonstrating the effectiveness of LanguageBind.
- **[Reviewer ePEX]** We clarified the impact of different textual sources on various modalities.


- **[Reviewer Bowz & ZKDg]** We added Table 6 to showcase the learning of stronger representations using available modalities, such as detection in dark environments. We have also added Table 7 to demonstrate non-text retrieval tasks.
- **[Reviewer Bowz & ePEX]** We added Table 3 to clarify the effectiveness of the dataset and made modifications to Table 2 to demonstrate the effectiveness of our LanguageBind compared to InternVideo.

We sincerely hope that this work will shed some light on the field of multimodal learning. Once again, we thank all the reviewers for spending their valuable time to help improve our work.

---

### Meta-Review · Area_Chair_q3Fc · 2023-12-09

**Metareview:**

The paper address the problem of multimodal representation learning and has two main contributions. On one side, it proposes to align all modalities along the text modality, using the representation of a frozen language model. It demonstrates empirically the utility of this approach. On the other side, it creates a new dataset of 10M videos, a pretty substantial number, that have 5 different modalities for each video, which is also an valuable asset.

All reviewers appreciate the idea of aligning all modalities along text and the value of the dataset. The results presented in the paper are also quite convincing, e.g. cross model retrieval, multimodal zero-shot classification across various combinations of modalities.

Some of the concerns with the paper are the quality of the dataset, as some of the modalities are generated by an algorithm. Further, some of the reviewers have raised limited novelty as the approach is very similar to ImageBind.

**Justification For Why Not Higher Score:**

The paper has received 3 x borderline accept and 1 x accept. The reviewers have certain reservations w.r.t limited novelty and the quality of the dataset.

**Justification For Why Not Lower Score:**

All reviewers have indicated desire to see the paper accepted. In particular, the paper presents strong empirical justification of an idea, that has a variation of a published work. Further, the paper comes with a novel, large, and multimodal dataset that can be quite useful to the community.

---

### Decision · Program_Chairs · 2024-01-16

Accept (poster)